# On the Evolution of the Hubble Constant with the SNe Ia Pantheon Sample and Baryon Acoustic Oscillations: A Feasibility Study for GRB-Cosmology in 2030

**Maria Giovanna Dainotti** [1,2,3,*] **, Biagio De Simone** [4,5] **, Tiziano Schiavone** [6,7] **, Giovanni Montani** [8,9] **, Enrico Rinaldi** [10,11,12] **, Gaetano Lambiase** [4,5] **, Malgorzata Bogdan** [13,14] **and Sahil Ugale** [15]

1. National Astronomical Observatory of Japan, 2 Chome-21-1 Osawa, Mitaka, Tokyo 181-8588, Japan
2. School of Physical Sciences, The Graduate University for Advanced Studies, Shonankokusaimura, Hayama, Miura District, Kanagawa 240-0193, Japan
3. Space Science Institute, Boulder, CO 80301, USA
4. Department of Physics "E.R. Caianiello", University of Salerno, Via Giovanni Paolo II, 132, Fisciano, I-84084 Salerno, Italy; bdesimone@unisa.it (B.D.S.); glambiase@unisa.it (G.L.)
5. INFN Gruppo Collegato di Salerno—Sezione di Napoli—c/o Dipartimento di Fisica "E.R. Caianiello", Ed. F, Università di Salerno—Via Giovanni Paolo II, 132, Fisciano, I-84084 Salerno, Italy
6. Department of Physics "E. Fermi", University of Pisa, Polo Fibonacci, Largo B. Pontecorvo 3, I-56127 Pisa, Italy; tiziano.schiavone@phd.unipi.it
7. INFN, Istituto Nazionale di Fisica Nucleare, Sezione di Pisa, Polo Fibonacci, Largo B. Pontecorvo 3, I-56127 Pisa, Italy
8. ENEA, Fusion and Nuclear Safety Department, C.R. Frascati, Via E. Fermi 45, Frascati, I-00044 Rome, Italy; giovanni.montani@enea.it
9. Physics Department, "Sapienza" University of Rome, P.le Aldo Moro 5, I-00185 Rome, Italy
10. Physics Department, University of Michigan, Ann Arbor, MI 48109, USA; erinaldi.work@gmail.com
11. Theoretical Quantum Physics Laboratory, Center for Pioneering Research, RIKEN, 2-1 Hirosawa, Wako, Saitama 351-0198, Japan
12. Interdisciplinary Theoretical & Mathematical Science Program, RIKEN (iTHEMS), 2-1 Hirosawa, Wako, Saitama 351-0198, Japan
13. Department of Mathematics, University of Wroclaw, plac Uniwersytecki 1, 50-137 Wrocław, Poland; malgorzata.bogdan@uwr.edu.pl
14. Department of Statistics, Lund University, P.O. Box 117, SE-221 00 Lund, Sweden
15. Department of Physics, Mithibai College, Mumbai 400056, India; sahil.ugale@svkmmumbai.onmicrosoft.com
* Correspondence: maria.dainotti@nao.ac.jp

**Abstract:** The difference from 4 to 6 $\sigma$ in the Hubble constant ($H_0$) between the values observed with the local (Cepheids and Supernovae Ia, SNe Ia) and the high-z probes (Cosmic Microwave Background obtained by the Planck data) still challenges the astrophysics and cosmology community. Previous analysis has shown that there is an evolution in the Hubble constant that scales as $f(z) = \mathcal{H}_0/(1+z)^\eta$, where $\mathcal{H}_0$ is $H_0(z=0)$ and $\eta$ is the evolutionary parameter. Here, we investigate if this evolution still holds by using the SNe Ia gathered in the Pantheon sample and the Baryon Acoustic Oscillations. We assume $H_0 = 70$ km s$^{-1}$ Mpc$^{-1}$ as the local value and divide the Pantheon into three bins ordered in increasing values of redshift. Similar to our previous analysis but varying two cosmological parameters contemporaneously ($H_0$, $\Omega_{0m}$ in the $\Lambda$CDM model and $H_0$, $w_a$ in the $w_0 w_a$CDM model), for each bin we implement a Markov-Chain Monte Carlo analysis (MCMC) obtaining the value of $H_0$ assuming Gaussian priors to restrict the parameters spaces to values we expect from our prior knowledge of the current cosmological models and to avoid phantom Dark Energy models with $w < -1$. Subsequently, the values of $H_0$ are fitted with the model $f(z)$. Our results show that a decreasing trend with $\eta \sim 10^{-2}$ is still visible in this sample. The $\eta$ coefficient reaches zero in 2.0 $\sigma$ for the $\Lambda$CDM model up to 5.8 $\sigma$ for $w_0 w_a$CDM model. This trend, if not due to statistical fluctuations, could be explained through a hidden astrophysical bias, such as the effect of stretch evolution, or it requires new theoretical models, a possible proposition is the modified gravity theories, $f(R)$. This analysis is meant to further cast light on the evolution of $H_0$ and it does not specifically focus on constraining the other parameters. This work is also a preparatory to understand how the combined probes still show an evolution of the $H_0$ by redshift and what is the current

status of simulations on GRB cosmology to obtain the uncertainties on the $\Omega_{0m}$ comparable with the ones achieved through SNe Ia.

**Keywords:** supernovae; Ia; cosmology; Hubble; tension; ΛCDM; evolution; modified; gravity; theories

## 1. Introduction

The ΛCDM model is one of the most accredited models, which implies an accelerated expansion phase [1,2]. Although it represents the favored paradigm, it is affected by great challenges: the fine-tuning, the coincidence [3,4], and the Dark Energy nature's problems.

More importantly, the $H_0$ tension represents a big challenge for modern cosmology. Indeed, the 4.4 up to 6.2 $\sigma$ discrepancy, depending on the sample used [5–7], between the local value of $H_0$ obtained with Cepheids observations and SNe Ia, $H_0 = 74.03 \pm 1.42 \, \text{km s}^{-1} \, \text{Mpc}^{-1}$ [8,9], and the Planck data of Cosmic Microwave background radiation (CMB), $H_0 = 67.4 \pm 0.5 \, \text{km s}^{-1} \, \text{Mpc}^{-1}$ from the Planck Collaboration [10] requires further investigation. From now on, $H_0$ will be in the units $\text{km s}^{-1} \, \text{Mpc}^{-1}$.

We stress that other probes report values of $H_0 \approx 72 \pm 2$, similar to the value obtained with the SNe Ia. Surely, to solve the Hubble tension it is necessary to use probes that are standard candles. SNe Ia, considered one of the best standard candles, are observed only up to a low redshift range: the farthest so far discovered is at $z = 2.26$ [11].

It is important for studying the evolution of the cosmological parameters to investigate probes at high redshift. One of the best candidates in this regard is represented by the Gamma-ray Bursts (GRBs).

GRBs are observed up to cosmological redshifts (the actual record is of $z = 9.4$ [12]) and surpassed even the quasars (the most distant being at $z = 7.64$ [13]). Due to their detectability at high redshift, GRBs allow extending the current Hubble diagram to new redshift ranges [14–18].

Indeed, it is important to stress that once we have established if the Hubble constant undergoes redshift evolution, the Pantheon sample can safely be combined with other probes. Surely the advantage of the use of the SNe Ia is that their emission mechanism is pretty clear, namely they originate from the thermonuclear explosion of carbon–oxygen white dwarfs (C/O WDs).

For GRBs, more investigation about their progenitor mechanism is needed. We here stress that this work can be also preparatory to the work of future application of GRBs as cosmological tools together with SNe Ia and Baryon Acoustic Oscillations (BAOs) through well-established correlations among the prompt variables, such as: the Amati relation [19], which connects the peak in the $\nu F_\nu$ spectrum to the isotropic energy in the prompt emission ($E_{iso}$), the Yonetoku relation [20,21] between $E_{peak}$ and the peak luminosity of the prompt emission, $L_{peak}$, the Liang and Zhang relation [22] between $E_{iso}$, the rest-frame break time of the GRB $t'_b$ and the peak energy spectrum in the rest frame $E'_p$, the Ghirlanda relation ($E_{peak} - E_{jet} = E_{iso} \times (1 - cos\theta)$) [23], and the prompt-afterglow relations for the GRBs with the plateau emission investigated in [24–38], which have as common emission mechanism most likely the magnetar model, where a neutron star with an intense magnetic field undergoes a fast-spinning down [39–43].

A feasibility study shows that GRBs can give relevant constraints on the cosmological parameters [17,44]. We here give a list of examples of other probes used for measuring the Hubble constant tension. One of them is the use of data from time-delay measurements and strong lens systems [45,46]. On the contrary, additional probes carry similar values of $H_0$ to the ones of Planck, based on the Cosmic Chronometers (CC) ($H_0 = 67.06 \pm 1.68$) in [8]. Besides, there is a series of independent probes, such as quasars [47], the Tip of the Red-Giant Branch (TRGB) calibration through SNe Ia [48], and also GRBs [14,15,17,18,49,50], which bring estimates of $H_0$ ranging between the values obtained with local measurements (SNe Ia and Cepheids) and Baryon Acoustic Oscillations (BAO)+CMB. Ref. [51] discuss possible

reasons behind the $H_0$ tension in the Pantheon sample: selection biases of parameters of the SNe Ia, unknown systematics, internal inconsistencies in the Planck data, or alternative theoretical interpretations compared to the standard cosmological model. Furthermore, the use of type 1 Active Galactic Nuclei (AGN) represents another promising cosmological probe given the peculiarity of their spectral emission [52].

To date, a wide range of different solutions to the Hubble constant tension has been provided by several groups [53–86]. Concerning the observational solutions, we here detail a series of proposals [87–130]. In [131], the simulations of data taken from the anomalously fast-colliding El Gordo galaxy clusters allow discussing the probability of observing such a scenario in a ΛCDM framework. Ref. [132] perform a re-calibration of Cepheids in NGC 5584, thus obtaining a relation between the periods of Cepheids and their amplitude ratios (tighter than the one obtained in SH0ES [9]) which could be useful to better estimate the value of $H_0$. In [133], the UV and X-ray data coming from quasars are used to constrain $H_0$ in the Finslerian cosmology. Ref. [123] demonstrate that the Planetary Nebula Luminosity Function (PNLF) can be extended beyond the Cepheid distances, thus promoting it to be an additive probe for constraining $H_0$. In [134], the analysis of Pan-STARRS telescope SNe Ia data provides a value of $H_0$ which lies between the SH0ES and Planck values.

Ref. [135] investigated how the $H_0$ measurements can depend on the choice of different probes (SNe, BAO, Cepheid, CC, etc.), showing also that through the set of filters on cosmological models, such as fiducial values for cosmological parameters ($w = -1$, with $w$ parameter for the equation of state, or $\Omega_k = 0$, namely the curvature parameter set to zero), the tension can be alleviated.

Ref. [136] extended a Hubble diagram up to redshift $z \sim 8$ combining galaxies and high-redshift quasars to test the late-time cosmic expansion history, giving a constraint on the upper-value of $H_0$ which is only marginally consistent with the results obtained by the Cepheids.

Ref. [137] further tests the $w$CDM (with varying parameters of the equation of state), and oCDM models (with varying curvature) through the merging of BAOs, SNe Ia, CC, GRBs, and quasars data, after the analysis of the standard ΛCDM model.

In [138], the combination of strongly lensed quasars and SNe Ia led the authors to conclude that the solution to the tension should be found outside of the Friedmann–Lemaitre–Robertson–Walker metric.

Refs. [125,139] detect in the Stochastic Gravitational Wave Background a new method to alleviate the tension, while [140,141] focuses on the gravitational-wave signals from compact star mergers as probes that can give constraints on the $H_0$ value.

Ref. [142] combine the SNe Ia and the VLT-KMOS HII galaxies data to put new constraints on the cosmokinetic parameters. The proposed solutions deal also with models that are alternative to the standard ΛCDM or, in other cases, that can extend it.

Ref. [143] constrain the Brans–Dicke (BD) theory through CMB and BAOs. The TRGB method, combined with SNe Ia, gives a value of $H_0$ compatible with the one from CMB [144].

Ref. [145] obtain an 8%-precise value of $H_0$ through the Fast Radio Bursts (FRB).

In [146], the Cepheids calibration parameters are allowed to vary, thus leading to an estimated value of $H_0$ which is compatible with the CMB one. The possibility that the Solar system's proper motion may induce a bias in the measurement of $H_0$ has been subject to study in [147], finding out that there is no degeneracy between the cosmological parameters and the parameters of the Solar system motion.

Ref. [148] measure $H_0$ through the galaxies parallax having as reference the CMB rest-frame, being this parallax caused by the peculiar motions.

Ref. [149] verified through the measurements on GRBs and quasars that the Hubble constant has a bigger value in the sky directions aligned with the CMB dipole polarization, suggesting that a detachment from the FLRW should be considered.

Refs. [150–153] investigate how the dark sirens producing gravitational waves could help to probe $H_0$. Despite being a promising method, the incompleteness of galaxy catalogs

may hinder the outcome of this method, thus [154] proposes a pixelated-sky approach to overcome the issue of event redshifts which are missing but may be retrieved through the galaxies present on the line of sight.

A review of the most promising emerging probes to measure the Hubble constant can be found in [155].

Recent results on the measurements of the Hubble parameter and constant through the Third LIGO-Virgo-KAGRA Gravitational-Wave Transient Catalog (GWTC-3) can be found in [156]. An evolving trend for $H_0$ may be naturally predicted in Teleparallelism [157–161], as well as in modified gravity theories [162–167]. Refs. [168–170] study the $f(Q, T)$ models in Teleparallel Gravity through CC and SNe Ia, thus obtaining a value of $H_0$ compatible in $1\,\sigma$ with the SH0ES result. The linear theory of perturbation for the $f(Q, T)$ theory is investigated in [171], allowing the future tests of this model through CMB data.

In [172], the Unimodular Gravity model is constrained with Planck 2018 [10], SH0ES, SNe Ia, and H0LiCOW collaboration [7]. Furthermore, the Axi–Higgs model is tested with CMB, BAO, Weak Lensing data (WL), and SNe [173]: in another paper, it is shown how this model relaxes the Hubble tension [173]. Ref. [174] describe the modified inflationary models considering constant-roll inflation. Ref. [175] give boundaries on the Hubble constant value with the gravitino mass conjecture.

Refs. [176–180] show the role of cosmological second-order perturbations of the flat $\Lambda$CDM model in the $H_0$ tension. Ref. [181] discuss how Dark Energy may be generated by quantum fluctuations of an inflating field and how the Hubble tension may be reduced by the spatial correlations induced by this effect. The Dark Energy itself may be subject to evolution, as pointed out in [182]. Ref. [183] show how a modification of the Friedmann equation may naturally explain the inconsistency between the local and the cosmological measurements of the Hubble constant.

Ref. [184] explain how the search for low-frequency gravitational waves (GWs) justifies the Hubble tension's solution through the assumption of neutrino-dark sector interactions. Ref. [185] show how the $R_K^{(*)}$ anomalies (namely, the discrepancy between the theoretical ratio of the fractions $B \to K^*\mu^+\mu^- / B \to K^*e^+e^-$ for the dilepton invariant mass bins from the Standard Model and the observed one, see [186]) and the $H_0$ tension can be solved by Dirac neutrinos in a two-Higgs-doublet theory.

The introduction of models where the cosmological axio-dilation is present may lead to a solution of the Hubble tension [187].

Refs. [188–191] discuss how the Early Dark Energy models (EDE) can be used to alleviate the $H_0$ tension. Ref. [192] analyze how the phantom Dark Energy models can give a limited reduction of the $H_0$ tension, while [193] explore how the Kaniadakis holographic Dark Energy model alleviates the $H_0$ tension.

In [194], the Viscous Generalized Chaplygin Gas (VGCG) model is used to diminish the Hubble tension. The holographic Dark Energy models are pointed as a possible solution through the study of unparticle cosmology [195].

Ref. [196] test seven cosmological models through the constraints of SNe Ia, BAO, CMB, Planck lensing, and Cosmic Chronometers with the outcome that in the $\Lambda$CDM scenario a flat universe is favored.

Ref. [197] discuss how the new physical scenarios before the recombination epoch imply the shift of cosmological parameters and how these shifts are related to the discrepancy between the local and non-local values of $H_0$.

Ref. [198] proposes that the $H_0$ tension may be solved if the speed of light is treated as a function of the scale factor (as in [199]), and applies this scenario to SNe Ia data.

Refs. [200,201] discuss the implementation of the alternative Phenomenologically Emergent Dark Energy model (PEDE), which can be also extended to a Generalized Emergent Dark Energy model (GEDE) with the addition of an extra free parameter. This shows the possibility of obtaining the PEDE or the $\Lambda$CDM cosmology as sub-cases of the GEDE scenario.

Ref. [202] consider a scenario of modified gravity predicting the increase of the expansion rate in the late-universe, thus proving that in this scenario the Hubble tension reduces significantly.

Ref. [203] study the $\Lambda$CDM model constrained, at the early-time universe, by the presence of the early Integrated Sachs–Wolfe (eISW) effect, proving that the early-time models aimed at attenuating the Hubble tension should be able to reproduce the same eISW effect just like the $\Lambda$CDM does. The observations of a locally higher value for $H_0$ led to the discussion of local measurements, constraints, and modeling. In this regards, the assumption of a local void [204–206] may produce locally an increased value for $H_0$.

The Universe appears locally inhomogeneous below a scale of roughly 100 Mpc. The question some cosmologists are attempting to solve is whether local inhomogeneities have impact on cosmological measurements and the Hubble diagram. Many observables are related to photons paths, which may be directly affected by the matter distribution. Many theoretical attempts were made during the last few decades to develop the necessary average prescription to evaluate the photon propagation on the observer's past light cone based on covariant and gauge-invariant observables [207–210]. Local inhomogeneities and cosmic structure cause scattering and bias effects in the Hubble diagram, which are due to peculiar velocities, selection effects, and gravitational lensing, but also to non-linear relativistic corrections [210–212]. This question was addressed in [213] utilising the N-body simulation of cosmic structure formation through the numerical code *gevolution*. This non-perturbative approach pointed out discrepancies in the luminosity distance between a homogeneous and inhomogeneous scenario, showing, in particular, the presence of non-Gaussian effects at higher redshifts. These studies related to distance indicators will become even more significant considering the large number of the forthcoming surveys designed to the observations on the Large Scale Structure of the Universe in the next decade (for instance, the Euclid survey [214,215] and the Vera C. Rubin Observatory's LSST [216]). The effect of local structures in an inhomogeneous universe should be considered in the locally measured value of $H_0$ [217,218]. The local under-density interpretation was also studied in Milgromian dynamics [131,219,220], but in [221] it is shown how this interpretation does not solve the tension. Ref. [219] study the KBC local void which is in contrast with the $\Lambda$CDM, thus proposing the Milgromian dynamics as an alternative to standard cosmology. Milgromian dynamics are studied also in [222] where, through the galactic structures and clusters, it is shown how this model can be consistent at different scales and alleviate the Hubble tension.

Ref. [223] describe the late time approaches and their effect on the Hubble parameter. The bulk viscosity of the universe is also considered the link between the early and late universe values of $H_0$ [224].

Ref. [225] explain how the local measurements over-constrain the cosmological models and propose the graphical analysis of the impact that these constraints have on the $H_0$ estimation through ad hoc triangular plots.

Refs. [220,226,227] describes the effects of inhomogeneities at small scales in the baryon density. Ref. [228] find out that the late time modifications can solve the tension between the $H_0$ SH0ES and CMB values through a parametrization of the comoving distance.

Ref. [229] propose to alleviate the Hubble tension considering an abrupt modification of the effective gravitational constant at redshift $z \approx 0.01$.

Other proposals are focused on the existence of different approaches.

Refs. [230,231] show how $H_0$ evolves with redshift at local scales.

Ref. [232] discuss how the breakdown of Friedmann–Lemaitre–Robertson–Walker (FLRW [233]) may be a plausible assumption to alleviate the Hubble tension. Ref. [234] investigate the binary neutron stars mergers and, with the analysis of simulated catalogs, show their potential to help to alleviate the $H_0$ tension.

Ref. [235] explain how Gaussian process (GP) and locally weighted scatter plot smoothing are used in conjunction with simulation and extrapolation (LOESS-Simex) methods can reproduce different sets of data with a high level of precision, thus giving new perspectives

on the Hubble tension through the simulation of Cosmic Chronometers, SNe Ia, and BAOs data sets.

Ref. [236] focus on the GP and state the necessity of lower and upper bounds on the hyperparameters to obtain a reliable estimation of $H_0$.

On the other hand, Ref. [237] suggested a novel approach to measure $H_0$ based on the distance duality relation, namely a method that connects the luminosity distance of a source to its angular diameter. In this case, data do not require a calibration phase and the relative constraints are not dependent on the underlying cosmological model.

Ref. [238] showed how the tension can be solved with a modified weak-field General Relativity theory, thus defining a local $H_0$ and a global $H_0$ value.

Ref. [239] investigated how a specific Dark Energy model in the generalized Proca theory can alleviate the tension.

In [240], the Horndeski model can describe with significantly good precision the late expansion of the universe thanks to the Hubble parameter data. The same model is considered promising for the solution of the $H_0$ tension in [241].

Ref. [242] described how the transition observed in Tully–Fisher data could imply an evolving gravitational strength and explain the tension.

Ref. [197] explain how the physical models of the pre-recombination era could cause the observed $H_0$ values discrepancy and suggest that if the local $H_0$ measurements are consistent then a scale-invariant Harrison–Zeldovich spectrum should be considered to solve the $H_0$ issue. The Dynamical Dark Energy (DDE) models are the object of study in [243,244]: in the former, the DDE is proposed as an alternative to $\Lambda$CDM, while in the latter it is shown how the Chevallier–Polarski–Linder (CPL) parametrization [245,246] is insensitive to Dark Energy at low redshift scales.

Refs. [247,248] propose Dark Radiation as a new surrogate of the Standard Model.

In [249], the scalar field cosmological model is used, together with the parametrization of the equation of state, to obtain $H_0$ and investigate the nature of Dark Energy. The possibility of a scalar field non minimally coupled to gravity as a probable solution to the $H_0$ tension is investigated in [250].

Ref. [251] highlight the advantage of the braneworld models to predict the local higher values of $H_0$ and, contemporaneously, respect the CMB constraints. Another approach is to solve the $H_0$ tension by allowing variations in the fundamental constants [252].

Ref. [253] propose a non-singular Einstein–Cartan cosmological model with a simple parametrization of spacetime torsion to alleviate the tension, while [254] propose a model where the Dark Matter is annihilated to produce Dark Radiation.

Ref. [255] introduce a hidden sector of atomic Dark Matter in a realistic model that avoids the fine-tuning problem. The observed weak effect of primordial magnetic fields can create clustering at small scales for baryons and this could explain the $H_0$ tension [256].

Ref. [257] test the General Relativity at galactic scales through Strong Gravitational Lensing. The Strong Lensing is a promising probe for obtaining new constraints on $H_0$, thanks to the next generation DECIGO and B-DECIGO space interferometers [258].

In [259], the cosmological constant $\Lambda$ is considered a dynamical quantity in the context of the running vacuum models and this assumption could tackle the $H_0$ tension. Ref. [260] show the singlet Majoron model to explain the acceleration of the expansion at later times and prove that this is consistent with large-scale data: this model has been subsequently discussed in other works [261]. The vacuum energy density value is affected by the Hubble tension as well and its measurement may cast more light on this topic [262].

Ref. [263] discuss the outcomes of the Oscillatory Tracker Model with an $H_0$ value that agrees with the CMB measurements. In [264], it is explained how the Generalized Uncertainty Principle and the Extended Uncertainty Principle can modify the Hubble parameter. [265] explore the implication of the Mirror Twin Higgs model and the need for future measurements to alleviate the tension.

The artificial neural networks can be applied to reconstruct the behavior of large scale structure cosmological parameters [266].

Another alternative is given by the gravitational transitions at low redshift which can solve the $H_0$ tension better than the late-time $H(z)$ smooth deformations [84,229].

Another comparison between the late-time gravitational transition models and other models which predict a smooth deformation of the Hubble parameter can be found in [267].

Ref. [268], as modifications to the $\Lambda$CDM model, consider as plausible scenarios or a Dark Matter component with negative pressure or the decay of Dark Energy into Dark Matter.

Ref. [269] does not observe the $H_0$ tension through the Effective Field Theory of Large Scale Structure and the Baryon Oscillation Spectroscopic Survey (BOSS) Correlation Function.

Considering the Dark Matter particles with two new charges, Ref. [270] reproduce a repulsive force which has similar effects to the $\Lambda$ cosmological constant. Furthermore, the models where interaction between Dark Matter and Dark Energy is present are promising for a solution of the Hubble constant tensions, see [271].

In [272], it is shown how two independent sets of cosmological parameters, the background (geometrical) and the matter density (growth) component parameters, respectively, give consistent results and how the preference for high values of $H_0$ is less significant in their analysis.

Ref. [273] introduce a global parametrization based on the cosmic age which rules out the early-time and the late-time frameworks.

Ref. [274] point out, through the use of non-parametric methods, how the cosmological models may induce biases in the cosmological parameters. In the same way, the statistical analysis of galaxies' redshift value and distance estimations may be affected by biases which could, in turn, affect the estimation of $H_0$.

Ref. [275]. This consideration holds also for the quadruply lensed quasars which are another method to measure $H_0$ [276].

Ref. [277] use the machine learning techniques to measure time delays in lensed SNe Ia, these being an independent method to measure $H_0$.

Additionally, in [231] it is explained how an evolution of $H_0$ with the redshift is to be expected. If a statistical approach on the different $H_0$ values is used instead, together with the assumption of an alternative cosmology, another solution to the tension could be naturally implied [278].

Ref. [279] use data to reconstruct the $f(T)$ gravity function without assuming any cosmological model: this $f(T)$ could in turn represent a solution to the $H_0$ tension.

Ref. [124] discuss how the addition of scalar fields with particle physics motivation to the cosmological model which predicts Dark Matter can retrieve the observed abundances of the Big Bang Nucleosynthesis.

In [280], a Dark Matter production mechanism is proposed to alleviate the $H_0$ tension. A general review of the perspectives and proposals concerning the $H_0$ tension can be found in [281–283].

SNe Ia represents a very good example of standard candles. Here we consider also the contribution of geometrical probes, the so-called *standard rulers*: while standard candles show a constant intrinsic luminosity (or obey an intrinsic relation between their luminosity and other physical parameters independent of luminosity), standard rulers are characterized by a typical scale dimension. This property allows estimating their distance according to the apparent angular size. Among the possible standard rulers, the BAOs assume great importance for cosmological purposes.

We here investigate the $H_0$ tension in the Pantheon sample (hereafter PS) from [284] and we add the contribution of BAOs to the cosmological computations to check if the trend of $H_0$ found in [36] is present also with the addition of other probes. We here point out that the current analysis is not meant to constrain $\Omega_{0m}$ or any other cosmological parameters, but it is focused to study the reliability of the trend of $H_0$ as a function of the redshift.

We here point out that this analysis is not meant to constrain $\Omega_{0m}$ or any other cosmological parameters, but it is focused to study the reliability of the trend of $H_0$ as a

function of the redshift. The range of redshift in the PS goes from $z = 0.01$ to $z = 2.26$. We tackle the problem with a redshift binning approach of $H_0$, the same used in [51], but here we adopt a starting value of $H_0 = 70$ instead of 73.5: if a trend with redshift exists, it should be independent on the initial value for $H_0$. The systematic contributions for the PS are calibrated through a reference cosmological model, where $H_0$ is 70.0 [284]. In the current paper, the aforementioned systematic uncertainties are considered for the analysis. Our approach has a two-fold advantage: on the one hand, it is relatively simple and on the other hand, it avoids the re-estimation of the SNe Ia uncertainties and may be able to highlight a residual dependence on the SNe Ia parameters with redshift.

While a slow varying Einstein constant with the redshift, as it emerges in a modified $f(R)$ gravity, appears as the most natural explanation for a trend $H_0(z)$, the analysis of Section 7 seems to indicate that such effect is not necessarily related with the Dark Energy contribution of the late universe. Since the Hu–Sawicki gravity lacks of reproducing the correct profile $H_0(z)$ shows that a Dark Energy model in the late Universe may not be enough to explain the observed effect since the scalar mode dynamics can not easily conciliate the Dark Energy contribution with the decreasing trend of $H_0(z)$. Thus, it may be necessary a modified gravity scenario more general than a Dark Energy model in the late Universe.

The current paper is composed as expressed in the following: in Section 2 the $\Lambda$CDM and $w_0 w_a$CDM models are briefly introduced together with SNe Ia properties; Section 3 describes the use of BAOs as cosmological rulers; Section 4 contains our binned analysis results, after slicing the PS in 3 redshift bins for the aforementioned models, and assuming locally $H_0 = 70$; in Section 5, we investigate, through simulated events, how the GRBs will be contributing to cosmological investigations by 2030; in Section 6 we discuss the results; in Section 7 we test the Hu–Sawicki model through a binning approach; in Section 8 we report an overview on the requirements that a suitable $f(R)$ model should have to properly describe the observed trend of $H_0$ and in Section 9 our conclusions are reported.

## 2. SNe Ia Cosmology

SNe Ia are characterized by an intrinsic luminosity that is almost uniform. Because of this, SNe Ia are considered reliable *standard candles*. We compare the theoretical distance moduli $\mu_{th}$ with the observed distance moduli $\mu_{obs}$ of SNe Ia belonging to the PS. The theoretical distance moduli are defined through the luminosity distance $d_L(z)$ which we need to define based on the cosmological model of interest. We here show the CPL parametrization which describes the $w$ parameter as a function of redshift ($w(z) = w_0 + w_a \times z/(1+z)$) in the $w_0 w_a$CDM model. In the usual assumptions $w_0 \sim -1$ and $w_a \sim 0$, and $d_L(z)$ is defined as the following [285]:

$$d_L(z, H_0...) = \frac{c(1+z)}{H_0} \int_0^z \frac{dz^*}{\sqrt{\Omega_{0m}(1+z^*)^3 + \Omega_{0\Lambda}(1+z^*)^{3(w_0+w_a+1)} e^{-3 w_a \frac{z^*}{1+z^*}}}}, \quad (1)$$

where $\Omega_{0\Lambda}$ is the Dark Energy component, $c$ is the speed of light, and $z$ is the redshift. We stress that in this context the relativistic components are ignored. Moreover, since in the present universe the radiation density parameter $\Omega_{0r} \approx 10^{-5}$, this contribution can be neglected. If we substitute $w_a = 0$, $w_0 = -1$ in Equation (1) the luminosity distance expression for $\Lambda$CDM model is automatically retrieved. According to the distance luminosity expression, the theoretical distance modulus can be written in the following form:

$$\mu_{th} = 5 \log_{10} d_L(z, H_0, \ldots) + 25, \quad (2)$$

which is usually expressed in Megaparsec (Mpc). The observed distance modulus, $\mu_{obs} = m'_B - M$, taken from PS contains the apparent magnitude in the B-band corrected for statistical and systematic effects ($m'_B$) and the absolute in the B-band for a fiducial SN Ia with a null value of stretch and color corrections ($M$). Considering the color and stretch population models for SNe Ia, in our approach we average the distance moduli given by

the [286] (G2010) and [287] (C2011) models. We here remind the reader that $H_0$ and $M$ are degenerate parameters: in the PS release, $M = -19.35$ such that $H_0 = 70.0$.

Ref. [51] obtain information on $H_0$ by comparing $\mu_{obs}$ in [284][1] with $\mu_{th}$ for each SN. Moreover, they fix $\Omega_{0m}$ to a fiducial value to better constrain the $H_0$ parameter. Furthermore, according to [288], we consider the correction of the luminosity distance keeping into account the peculiar velocities of the host galaxies which contain the SNe Ia. To perform our analysis, we define the $\chi^2$ for SNe:

$$\chi^2_{SN} = \Delta\mu^T \cdot \mathcal{C}^{-1} \cdot \Delta\mu. \tag{3}$$

Here $\Delta\mu = \mu_{obs} - \mu_{th}$, and $\mathcal{C}$ denotes the $1048 \times 1048$ *covariance matrix*, given by [284]. As for the $\mu_{obs}$ values of G2010 and C2011, the systematic uncertainty matrices of the two models have been averaged. After building the $\mathcal{C}$ total matrix from Equation (16) in [51], we slice the PS in redshift bins, and then we divide $\mathcal{C}$ into submatrices considering the order in redshift. More in detail, starting from the 1048 SNe Ia redshift-ordering, we divide the SNe Ia into 3 equally populated bins made up of $\approx$349 SNe Ia. Concerning only $D_{stat}$, it is trivial to build its submatrices considering that the statistical matrix is diagonal. Hence, a single matrix element is related to a given SN of the PS. On the other hand, if the non-diagonal matrix $C_{sys}$ is included, a customized code will be used[2] to build the submatrices. Our code was developed to select only the total covariance matrix elements related to SNe Ia having redshift within the considered bin.

The choice of three bins is justified by the high number of SNe Ia (around hundreds of SNe per bin) that can still constitute statistically illustrative subsamples of the PS and that can properly consider the contribution of systematic uncertainties. Subsequently to the bins division, we focus on the optimal values of $H_0$ to minimize the $\chi^2$ in Equation (3). $H_0$ is regarded as a nuisance parameter, which is free to vary, to better analyze a possible redshift function of $H_0$. We follow the assumptions on the fiducial value of $M = -19.35$: while in [51] $M$ was estimated assuming a local ($z = 0$) value of $H_0 = 73.5$, we here consider the conventional $H_0$ value of the PS release, namely $H_0 = 70.0$ for three bins. Our choice of a starting value of $H_0 = 70$ is dictated by the presence in the current literature of more than 50 papers that are using the PS in combination with other probes to estimate the value of $H_0$, see [172,198,204,205,288–307,307–339]. Thus, if an evolutionary effect is present, it is necessary to investigate to which extent this can affect current and future results largely based on the PS sample. Conversely, we fix $\Omega_{0m} = 0.298 \pm 0.022$ according to [284] for a standard flat $\Lambda$CDM model. More specifically, after the minimization of $\chi^2$, we extract the $H_0$ value in each redshift bin, via the *Cobaya* code [340]. To this end, we execute an MCMC using the D'Agostini method to obtain the confidence intervals for $H_0$ at the 68% and 95% levels, in three bins.

## 3. The Contribution of BAOs

The environment of relativistic plasma in the early universe was crossed by the sound waves that were generated by cosmological perturbations. At redshift $z_d \sim 1059.3$, which marks the ending of the drag period [341], the recombination of electrons and protons into a neutral gas interrupted the propagation of the sound waves while the photons were able to propagate further [342]. In the period between the formation of the perturbations and the recombination, the different modes produced a sequence of peaks and minima in the anisotropy power spectrum. Given the huge fraction of baryons in the universe, it is expected by cosmological models that the oscillations may affect also the distribution of baryons in the late universe. As a consequence, the BAOs manifest as a local maximum in the correlation function of the galaxies distribution in correspondence of the comoving sound horizon scale at the given redshift $z_d$, namely $r_s(z_d)$: this is associated with the stopping of the propagation of the acoustic waves.

To use the BAOs data for cosmology, we first need to define the following variables:

$$D_V(z) = \left[ \frac{czd_L^2(z)}{(1+z)^2 H(z)} \right]^{1/3}, \qquad d_z(z) = \frac{r_s(z_d)}{D_V(z)}. \tag{4}$$

The value of the redshift $z_d$, which corresponds to the drag era ending and marks the decoupling of the photons, allows estimating the sound horizon scale:

$$(r_d \cdot h)_{fid} = 104.57\,\text{Mpc}, \qquad r_s(z_d) = \frac{(r_d \cdot h)_{fid}}{h}, \tag{5}$$

where we use the adimensional ratio $h = H_0/100(\text{km s}^{-1}\ \text{Mpc}^{-1})$. To estimate $r_s$, the following approximated formula [343] can be applied:

$$r_s \approx \frac{55.154 \cdot e^{-72.3(\omega_v + 0.0006)^2}}{\omega_{0m}^{0.25351}\omega_b^{0.12807}}\,\text{Mpc}, \tag{6}$$

where $\omega_i = \Omega_i \cdot h^2$, and $i = m, v, b$ represent matter, neutrino and baryons. We here assume $\omega_v = 0.00064$ [344] and $\omega_b = 0.02237$ [10]. Given these quantities, we define the $\chi^2$ for BAOs as follows:

$$\chi^2_{BAO} = \Delta d^T \cdot \mathcal{M}^{-1} \cdot \Delta d, \tag{7}$$

where $\Delta d = d_z^{obs}(z_i) - d_z^{theo}(z_i)$ and $\mathcal{M}$ is the covariance matrix for the BAO $d_z^{obs}(z_i)$ values. In this binned analysis, a subset of the 26 BAO observations set available in [341] will be employed.

## 4. Multidimensional Binned Analysis with SNe Ia and BAOs

To investigate the $H_0$ tension through the SNe Ia and BAOs data, we combine the $\chi^2$ Equations (3) and (7) to obtain the total $\chi^2$

$$\chi^2 = \frac{1}{2}\chi^2_{SN} + \frac{1}{2}\chi^2_{BAO}, \tag{8}$$

In our work, we combine each SNe bin with only 1 BAO data point which has a redshift value within the SNe bin: this approach of using one BAO comes from [18]. In this way we do not have the problem of a different number of BAOs in different bins. Through Equation (8), we investigate if a redshift evolution of $H_0(z)$ is present, obtaining it from the binning of SNe Ia+BAOs considering three bins with the $\Lambda$CDM and $w_0 w_a$CDM models. A feasibility study done in [51] performed with different bins selections has highlighted how the maximum number of bins in which the PS should be divided is 3, otherwise the statistical fluctuations would dominate on a multi-dimensional analysis, leading to relatively large uncertainties which would mask any evolving trend, if present. Furthermore, for the same reason, it is not advisable to leave free to vary more than two parameters at the same time, thus in the current section, we will analyze the behavior of $H_0$ in three bins when it is varied together with a second cosmological parameter. The same considerations make necessary the choice of more tight priors since we are basing the current analysis on the prior knowledge, avoiding the degeneracies among the parameter space, and letting the priors have more weight in the process of posteriors estimation. Differently from [51], for the $\Lambda$CDM model, we will let the parameters $H_0$ and $\Omega_{0m}$ vary simultaneously, while in the $w_0 w_a$CDM model the varying parameters are $H_0$ and $w_a$. We decided to leave $w_a$ free to vary since, according to the CPL parametrization, $w_a$ gives direct information about the evolution of the $w(z)$ while $w_0$ is considered a constant in the same model. Concerning the fiducial values and the priors assignment for the MCMC computations, we apply Gaussian priors with mean equal to the central values of $\Omega_{0m} = 0.298 \pm 0.022$ and $H_0 = 70.393 \pm 1.079$ for $\Omega_{0m}$ and $H_0$, respectively, and with $1\,\sigma = 2*0.022$ and $1\,\sigma = 2*1.079$ for $\Omega_{0m}$ and $H_0$,

respectively. In summary, to draw the Gaussian priors, we consider the mean value of the parameters as the expected one of the Gaussian distribution and we double the $\sigma$ value which is then considered the new standard deviation for the distribution. Concerning the $w_0 w_a$CDM model, we fix $w_0 = -0.905$ and we consider the priors on $w_a$ with the mean = $-0.129$ taken from Table 13 of [284], while 1 $\sigma =$ is the 20% of its central value. Such an assumption with small prior is needed since we need to assume that $w(z) > -1.168$ as the value tabulated in [284]. Besides, since we are here dealing with standard cosmologies, with this constraint we are avoiding some of the phantom Dark Energy models.

After the $\chi^2$ minimization for each bin, we perform a MCMC simulation to draw the mean value of $H_0$ and its uncertainty. Once $H_0$ is obtained for each bin, we perform a fit of $H_0$ using a simple function largely employed to characterize the evolution of many astrophysical objects, such as GRBs and quasars [17,29,31,35,345–349]. More specifically, the fitting of $H_0$ is given by

$$f(z) = H_0(z) = \frac{\mathcal{H}_0}{(1+z)^\eta},\tag{9}$$

in which $\mathcal{H}_0$ and $\eta$ are the fitting parameters. The former $\mathcal{H}_0 \equiv H_0$ at $z = 0$, while the latter $\eta$ coefficient describes a possible evolutionary trend of $H_0$. We consider the 68% confidence interval at, namely 1 $\sigma$ uncertainty.

In the current treatment, we consider the calibration of the PS with $H_0 = 70$ as provided by [284]. Results are presented in the panels of Table 1. We here stress that the fiducial magnitude value is assumed to be $M = -19.35$ for each SNe bin, thus it will not be mentioned in the same Table. All the uncertainties in the tables in this paper are in 1 $\sigma$. As reported in the upper half of Table 1, namely with the $\Lambda$CDM model, if we do not include the BAOs then the $\eta$ coefficient is compatible with 0 in 2.0 $\sigma$ for the three bins case. When we introduce the BAOs within the $\Lambda$CDM model, we observe again a reduction of the $\eta/\sigma_\eta$ ratio for three bins down to 1.2. Concerning the lower half of Table 1 with the $w_0 w_a$CDM model, when BAOs are not included we have $\eta$ non compatible with 0 in 5.7 $\sigma$ and, including the BAOs, the compatibility with 0 is given in 5.8 $\sigma$. The increasing of the ratio $\eta/\sigma_\eta$ is observed when BAOs are added in the case of $w_0 w_a$CDM model in three bins. The results can be visualized in Figure 1. Comparing the $\eta/\sigma_\eta$ ratios with the ones reported in [51] (Table 1) we have that for the $\Lambda$CDM model the current $\eta$ values are compatible in 1 $\sigma$ with the $\alpha$ reported in [51], while the $\eta$ estimated in the $w_0 w_a$CDM model are compatible in 3 $\sigma$ with the $\alpha$ values in the same reference paper.

**Table 1. Upper half.** Fit parameters of $H_0(z)$ for three bins (flat $\Lambda$CDM model, varying $H_0$ and $\Omega_{0m}$) in the cases with SNe only and with the SNe + BAOs contribution. The columns are: (1) the number of bins; (2) $\mathcal{H}_0$, (3) $\eta$; (4) how many $\sigma$s the evolutionary parameter $\eta$ is compatible with zero (namely, $\eta/\sigma_\eta$). **Lower half.** Similarly to the upper half, the lower half shows the fit parameters of $H_0(z)$ (flat $w_0 w_a$CDM model, varying $H_0$ and $w_a$) without and with the BAOs.

| Flat $\Lambda$CDM model, without BAOs, varying $H_0$ and $\Omega_{0m}$ | | | |
|:---:|:---:|:---:|:---:|
| Bins | $\mathcal{H}_0$ | $\eta$ | $\frac{\eta}{\sigma_\eta}$ |
| 3 | $70.093 \pm 0.102$ | $0.009 \pm 0.004$ | 2.0 |
| Flat $\Lambda$CDM model, including BAOs, varying $H_0$ and $\Omega_{0m}$ | | | |
| Bins | $\mathcal{H}_0$ | $\eta$ | $\frac{\eta}{\sigma_\eta}$ |
| 3 | $70.084 \pm 0.148$ | $0.008 \pm 0.006$ | 1.2 |

**Table 1.** *Cont.*

| Bins | $\mathcal{H}_0$ | $\eta$ | $\frac{\eta}{\sigma_\eta}$ |
|---|---|---|---|
| Flat $w_0 w_a$CDM model, without BAOs, varying $H_0$ and $w_a$ | | | |
| 3 | $69.847 \pm 0.119$ | $0.034 \pm 0.006$ | 5.7 |
| Flat $w_0 w_a$CDM model, including BAOs, varying $H_0$ and $w_a$ | | | |
| 3 | $69.821 \pm 0.126$ | $0.033 \pm 0.005$ | 5.8 |

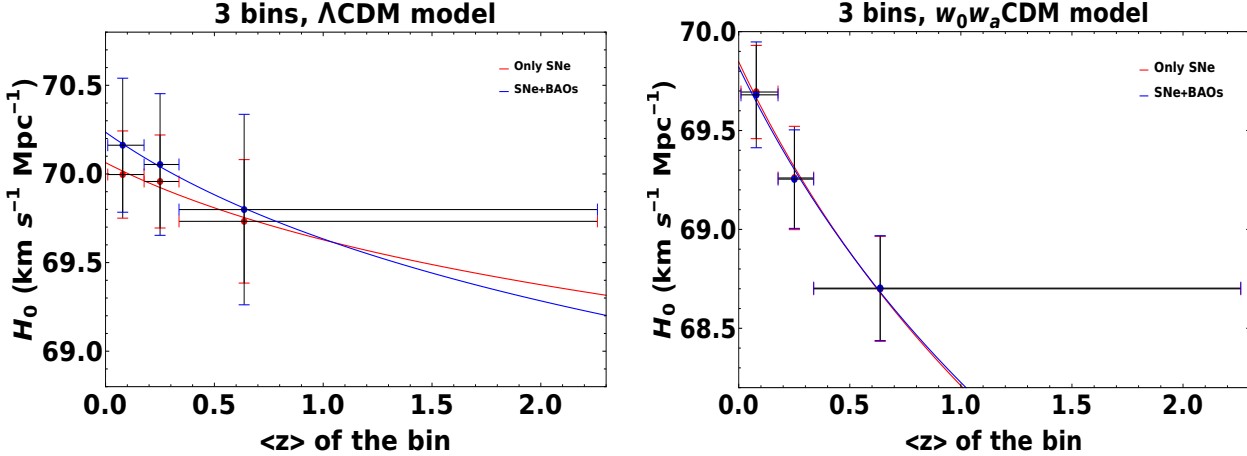

**Figure 1. Left panel** The $H_0(z)$ vs. z wit varying also $\Omega_0 M$. The red color indicates the case with only SNe Ia as probes, while the blue refers to the case of SNe + 1 BAO per bin. This color-coded will be applied also in the right panel. **Right.** The same plot for the $w_0 w_a$CDM model, considering the local fiducial value $H_0 = 70$, where both $H_0$ and $w_a$ are left free to vary.

## 5. Perspective of the Future Contribution of GRB-Cosmology in 2030

The discussion of GRBs as possible cosmological tools has been going on for more than two decades [350,351]. The best bet is yet to come since we need first to identify the tightest correlation possible with a solid physical grounding. Among the many correlations proposed [19–23] we here choose to apply the *fundamental plane* (or Dainotti relation) [30,352–354], namely the three-dimensional relation between the end of the plateau emission's luminosity, $L_a$, its time in the rest-frame, $T_a^*$, and the peak luminosity of the GRB, $L_{peak}$: it is possible to estimate how many GRBs are needed to obtain constraints for the cosmological parameters that are comparable with the ones obtained from the other probes, such as SNe Ia and BAOs. After a selection of the best fundamental plane sample through the trimming of GRBs, a simulation of a sample of 1500 and 2000 GRBs according to the properties of the fundamental plane relation has been performed. The fundamental plane relation can be expressed as the following:

$$log_{10}L_a = a \times log_{10}T_a^* + b \times log_{10}L_{peak} + c, \tag{10}$$

where $a, b$ are the parameters of the plane and $c$ is the normalization constant. It is important to stress that here the variables $L_a$, $T_a^*$, and $L_{peak}$ have been corrected for evolutionary effects with redshift applying the Efron and Petrosian method [355]. Based on Equation (10), we perform the maximization of the following log-likelihood for the simulated sample of GRBs:

$$
\begin{aligned}
ln\mathcal{L}_{GRB} = {} & -\frac{1}{2}\left(ln(\sigma^2 + (a * \delta_{log_{10}T_a^*})^2 + (b * \delta_{log_{10}L_{peak}})^2 + \delta^2_{log_{10}L_a})\right) \\
& -\frac{1}{2}\left(\frac{(log_{10}L_{a,th} - log_{10}L_a)^2}{\sigma^2 + (a * \delta_{log_{10}T_a^*})^2 + (b * \delta_{log_{10}L_{peak}})^2 + \delta^2_{log_{10}L_a}}\right),
\end{aligned}
\tag{11}
$$

where $L_{a,th}$ is the theoretical luminosity computed through the fundamental plane in Equation (10), $\sigma$ is the intrinsic scatter of the plane and $\delta_{log_{10}T_a^*}$, $\delta_{log_{10}L_{peak}}$, and $\delta_{log_{10}L_a}$ are the errors on the rest-frame time at the end of the plateau emission, the peak luminosity and the luminosity at the end of the plateau, respectively.

After performing an MCMC analysis using the D'Agostini method [356] and letting vary the parameters $a, b, c, \sigma, \Omega_{0m}$, the results are shown in Figure 2. Through the simulations of 1000 GRBs, with 9500 steps and keeping the same errors (errors undivided) as the ones observed in the fundamental plane ($n = 1$, see the upper left panel of Figure 2) we obtain a value of $\Omega_{0m} = 0.310$ with a symmetrized uncertainty of $\sigma_{\Omega_{0m}} = 0.078$. In the case of 2000 GRBs with 13,000 steps and $n = 1$ (see the upper right panel of Figure 2) instead, we have $\Omega_{0m} = 0.300$, $\sigma_{\Omega_{0m}} = 0.052$. If we consider the division of the errors on the variables of the fundamental plane by a factor 2 (halved errors, $n = 2$) we obtain, in the case of 1500 GRBs with 11,100 steps, $\Omega_{0m} = 0.300$, $\sigma_{\Omega_{0m}} = 0.037$ (see the lower-left panel of Figure 2), while through 2000 simulated GRBs in 14,600 steps (still with $n = 2$, see the lower right panel of Figure 2) we have $\Omega_{0m} = 0.310$, $\sigma_{\Omega_{0m}} = 0.034$. The idea of considering halved errors comes from the prospects for improvement in the fitting procedures of GRB light curves. Through this approach, the GRBs have provided constraints on the value of $\Omega_{0m}$ that are compatible with the ones of previous samples of SNe Ia: in the $n = 1$ cases, the values of the uncertainties are comparable with the ones from [357], while for the $n = 2$ cases the values are close to the ones found in [358] with 2000 GRBs. Furthermore, the GRBs have proven to be promising standardizable candles and, given the bigger redshift span they can cover if compared with SNe Ia, GRBs will provide more complete information about the structure and the evolution of the early universe after the Big Bang, together with quasars [359,360]. After discussing the potentiality of GRBs as future standard candles, we estimate the frequency of GRBs with a plateau emission over the total number of GRBs observed to date. We can expect that by 2030 we will have reached several GRB observations such that these—as standalone probes that respect the properties of the GRB platinum sample [35]—will give constraints as precise as the ones from [357] in the case of not halved errors. In case of halved errors, we can reach the level of precision of [357] even now. In addition, if we consider a machine learning analysis [361,362] for which we can double the size of the sample we are able to reach the precision of [357] now with the case of $n = 1$. If we consider the case of reconstructing the light curves and thus we have a sample which has the 47% of cases with halved errors we can reach the limit of [357] in 2022 if $n = 1$ and now if $n = 2$.

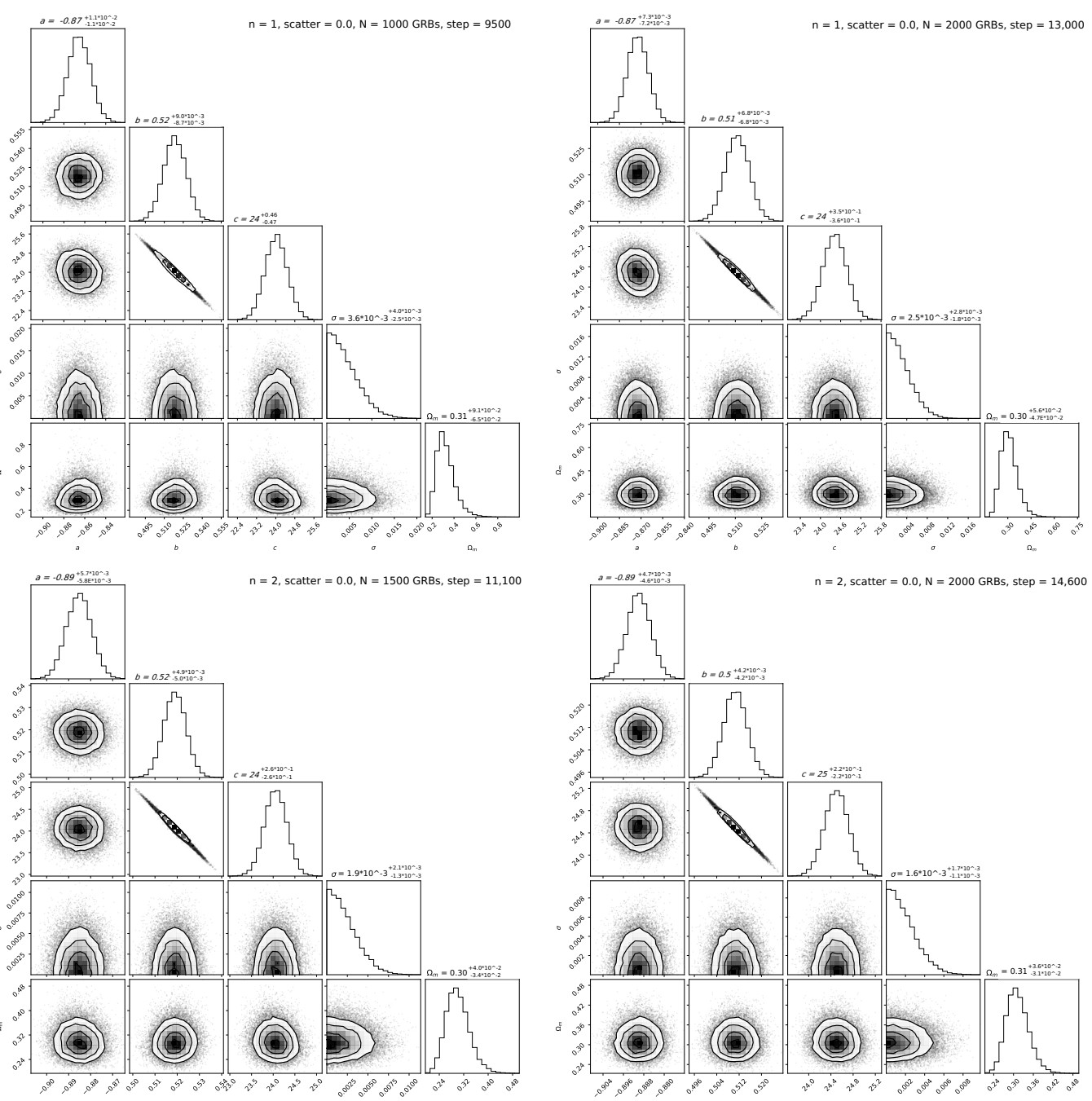

**Figure 2. Upper left.** An example of 1000 simulated GRBs with the posterior distribution of the fundamental plane parameters *a*, *b*, *c*, and its intrinsic scatter $\sigma$ together with the total matter density parameter $\Omega_{0m}$. In this case, the steps of the simulation are 9500 and the errors on the variables of the fundamental plane have not been divided by any factor ($n = 1$). **Upper right.** The same case of the upper left panel, but considering 2000 GRBs and a number of steps of 13,000. **Lower left.** The results of 1500 simulated GRBs, dividing by two the errors on the fundamental plane variables (halved errors, $n = 2$): here the steps are 11,100. **Lower right.** The same result of the lower-left panel, considering 2000 GRBs and 14,600 steps instead.

## 6. Discussions on the Results

Our results can be interpreted because of astrophysical selection biases or theoretical models alternatives to the standard cosmological models.

*6.1. Astrophysical Effects*

The main effect that has a stake in the SNe Ia luminosity variation is the presence of metallicity and the difference in stellar ages. Indeed, ref. [284] correct the PS with a mass-step contribution ($\Delta M$). Despite this term improving the results, other effects need to be accounted for. Considering the stretch and the color, ref. [363] claim that the Hubble residuals, after being properly corrected according to the stretch and color observations, for SNe Ia in low mass and high mass host galaxies show a difference of $0.077 \pm 0.014$ mag, compatibly with the result of [284]. SNe Ia age metallicity and age are believed to be responsible for the observed behavior: those can replicate the Hubble residual trends consistent with the ones of [363]. In the PS, to account for the evolutions of stretch ($\alpha$) and color ($\beta$), the parametrization utilized is the following: $\alpha(z) = \alpha_0 + (\alpha_1 \times z)$, $\beta(z) = \beta_0 + (\beta_1 \times z)$. According to [284], there is no clear dependence on the redshift for $\alpha(z)$ and $\beta(z)$, thus $\alpha_1$ and $\beta_1$ are set to zero. Only the selection effect for color is noteworthy and [284] consider the uncertainty on $\beta_1$ as a statistical contribution. Concerning the stretch evolution in the PS calibration, it appears to be negligible and is not included at any level.

Conversely, ref. [364] recently studied the SALT2.4 lightcurve stretch and showed that the SN stretch parameter is redshift-dependent. According to their analysis, the asymmetric Gaussian model assumed by [284] for describing the populations of SNe Ia does not take into consideration the redshift drift of the PS, thus leaving a residual evolutionary trend that manifests at higher redshifts. Indeed, the simulations performed by [284] for studying the systematics calibration reach redshifts up to $z = 0.7$: this threshold is present in the third bin of our analysis. The effect from $0.7 \leq z \leq 2.26$ needs additional investigations. It is worth noting that this decreasing trend of $H_0$ (with a given value of $\eta$) found in [51] is consistent in $1\,\sigma$ for the $\Lambda$CDM both in the cases of SNe Ia only and SNe+BAOs. When we consider the $w_0 w_a$CDM, the $\eta$ values are compatible in $3\,\sigma$ with the ones with SNe Ia only and SNe + BAOs. We here have two cosmological parameters varying at the same time, differently from [51]. Therefore, one of the possible astrophysical reasons behind the observed trend is the residual stretch evolution with redshift. If so, in our work the effect is simply switched from stretch to $H_0$. The forthcoming release of the Pantheon+ data [107,365–369] will give the chance to test if these evolutionary effects may be still visible, but this analysis goes far beyond the scope of the current paper. The astrophysical interpretation seems to be favored, but also many theoretical explanations may be possible to describe the outcome of these results.

*6.2. Theoretical Interpretations*

We now investigate possible theoretical explanations for our results, focusing particular attention on modified gravity models. We first discuss a general scalar-tensor formulation and, then, we concentrate our attention on the so-called metric $f(R)$ gravity.

6.2.1. The Scalar Tensor Theory of Gravity

The action of the scalar tensor theories (STTs) of gravity is given by $S = S^{JF} + S_m$ [370–374] with the Jordan Frame (JF) action

$$S^{JF} = \frac{1}{16\pi} \int d^4x \sqrt{-\tilde{g}} \left[ \Phi^2 \tilde{R} + 4\,\omega(\Phi)\tilde{g}^{\mu\nu}\partial_\mu\Phi\partial_\nu\Phi - 4\tilde{V}(\Phi) \right], \tag{12}$$

where $\tilde{R}$ is the Ricci scalar obtained with the physical metric $\tilde{g}_{\mu\nu}$, while the matter fields $\Psi_m$ couple to the metric tensor $\tilde{g}_{\mu\nu}$ and not to $\Phi$, i.e., $S_m = S_m[\Psi_m, \tilde{g}_{\mu\nu}]$.

In this Section we adopt natural units such that $c = 1$ and $G = 1$. Different STTs follow with the appropriate choice of the two functions $\omega(\Phi)$ and $\tilde{V}(\Phi)$: e.g., the Brans–Dicke (BD) theory [375–378] can be obtained for $\omega(\Phi) = \omega$ (const.) and $\tilde{V}(\Phi) = 0$, while the metric $f(R)$ gravity, discussed in the next subsection, would correspond to $\omega \equiv 0$.

The action $S^{JF}$ can be rewritten in the Einstein Frame (EF), where one defines $\tilde{g}_{\mu\nu} \equiv A^2(\varphi)g_{\mu\nu}$, $\Phi^2 \equiv 8\pi M_*^2 A^{-2}(\varphi)$, $V(\varphi) \equiv \frac{A^4(\varphi)}{4\pi}\tilde{V}(\Phi)$, $\gamma(\varphi) \equiv \frac{d\log A(\varphi)}{d\varphi}$, and $\gamma^2(\varphi) = \frac{1}{4\omega(\Phi)+6}$, to get

$$S^{EF} = \frac{M_*^2}{2}\int d^4x \sqrt{-g}\left[R + g^{\mu\nu}\partial_\mu\varphi\partial_\nu\varphi - \frac{2}{M_*^2}V(\varphi)\right]. \tag{13}$$

Matter is coupled to $\varphi$ only through a purely metric coupling, $S_m = S_m[\Psi_m, A^2(\varphi)g_{\mu\nu}]$ and $M_*$ is the Planck mass.

The physical quantities in the Jordan and Einstein frame are related by $d\tilde{\tau} = A(\varphi)d\tau$, $\tilde{a} = A(\varphi)a$, $\tilde{\rho} = A(\varphi)^{-4}\rho$, $\tilde{p} = A(\varphi)^{-4}p$, where $\tau$ is the synchronous time variable. Defining $N \equiv \log\frac{a}{a_0}$, $\lambda \equiv \frac{V(\varphi)}{\rho}$, $w \equiv \frac{p}{\rho}$, and $\varphi' \equiv \frac{d\varphi}{dN} = a\frac{d\varphi}{da}$, the combination of cosmological equations allows to write the equation for $\varphi$ in the form (for a flat Friedmann–Robertson–Walker geometry) [379]

$$\frac{2}{3}\frac{1+\lambda}{1-\varphi'^2/6}\,\varphi'' + [(1-w)+2\lambda]\varphi' = -\sqrt{2}\,\gamma(\varphi)\,(1-3w) - 2\,\lambda\,\frac{V_\varphi(\varphi)}{V}, \tag{14}$$

Moreover, the Jordan- and Einstein-frame Hubble parameters, $\tilde{H} \equiv d\log\tilde{a}/d\tilde{\tau}$ and $H \equiv d\log a/d\tau$, respectively, are related as

$$\tilde{H} = \frac{1+\gamma(\varphi)\,\varphi'}{A(\varphi)}\,H. \tag{15}$$

For our purpose, we consider $A(\varphi) = A_0 e^{c_1\varphi + c_2\varphi^2/2}$, which implies $\gamma(\varphi) = c_1 + c_2\varphi$, where $c_{1,2}$ are constants. Under the following conditions $\varphi''/\varphi \ll 1$, $\varphi'^2/\varphi^2 \ll 1$, and $\frac{V_\varphi(\varphi)}{\varphi V\rho} \ll 1$, the solution of Equation (14) is $\varphi(z) = C(1+z)^K - \frac{c_1}{c_2}$, where $K = \frac{1-3w}{1+w}\sqrt{2}c_2$, and $C$ is an integration constant. We are looking for solutions such that $H = f(\varphi)\tilde{H}_0$, so that $\tilde{H} = \frac{\tilde{H}_0}{(1+z)^\eta}$, where $\tilde{H}_0$ is constant. These relations and (15) allow to derive $f(\varphi)$ (the expression of $f(\varphi)$ is quite involved, and in the case in which $c_{1,2} \ll 1$, it is a polynomial in $\varphi$). The scalar field $\Phi$ in the (physical) JF can be cast in the form $\Phi(z) = \Phi_0(1+z)^{\tilde{K}}$, where $\Phi_0 \equiv \frac{\sqrt{8\pi}M_*}{A_0}\left[1 - C\left(c_1 - \frac{Cc_2}{c_1}\right)\right]$, $\tilde{K} = -\frac{KC(c_1+Cc_2)}{1-C\left(c_1+\frac{Cc_2}{2}\right)}$, and $z < 1$ has been used (note: $\tilde{K}$ is positive for $c_1$ or $c_2$ negative). The scalar field $\Phi$ reduces to $\phi$ for $\Phi_0 \to 1$ and $\tilde{K} \to 2\eta$. From the Friedmann Equation [379]

$$\left(\frac{\dot{a}}{a}\right)^2 = \frac{1}{3M_*^2}\left[\rho + \frac{M_*^2}{2}\dot{\varphi}^2 + V(\varphi)\right], \tag{16}$$

with $\rho$ given by matter ($\rho = \rho_{0m}/a^3 = \rho_{0m}(1+z)^3$), and $c_{1,2} \ll 1$, one infers the effective potential

$$\frac{\tilde{V}}{3m^2} = \frac{4\pi M_*^2}{A_0^2}\left[f_0^2 - \frac{1}{\Omega_{0m}}\left(\frac{\Phi}{\Phi_0}\right)^{\frac{3}{2\eta}} - \frac{C^2K^2\varphi_0^2}{6\Omega_{0m}}\left(\frac{\Phi}{\Phi_0}\right)^{\frac{K-\eta}{\eta}}\right], \tag{17}$$

where we recall that $\Omega_{0m} = \rho_{0m}/\rho_{cr}$, $\rho_{cr} = 3M_*^2\tilde{H}_0^2$, $f_0 = f(\varphi = 0)$, and $m^2 = \Omega_{0m}\tilde{H}_0^2$. For redshift $0 \leqslant z < 0.3$, to which we are interested, the scalar field varies slowly with $z$, $\Phi \sim \Phi_0$, so that the effective potential behaves like a cosmological constant. We see how the proposed scalar-tensor formulation has the right degrees of freedom to reproduce, in the JF, the required behavior of the (physical) trend of $H_0(z)$. In the next subsection, we analyze a sub-case of the general paradigm discussed above, which leads to the well-known $f(R)$ gravity, which is among the most popular modified gravity formulations.

6.2.2. Metric f(R) Gravity in the Jordan Frame

The observed decaying behavior of the Hubble constant $H_0$ with the redshift draws significant attention for an explanation and, if it is not due to selection effects or systematics in the sample data, we need to interpret our results from a physical point of view. As already argued in [51,380], the simplest way to account for this unexpected behavior of $H_0(z)$ is that the Einstein constant $\chi = 8\pi G$ (where $G$ denotes the gravitational constant), mediating the gravity-matter interaction, is subjected itself to a slow decaying profile with the redshift. In this Section, we consider $c = 1$ for the speed of light. More specifically, since the critical energy density $\rho_{c0} = 3H_0^2/\chi$ today must be a constant, we need an evolution for $\chi \sim (1+z)^{-2\eta}$, considering the function $H_0(z)$ given by Equation (9). The evolution of $\chi(z)$ is not expected within the cosmological Einsteinian gravity, therefore we are led to think of it as a pure dynamical effect, associated with a modified Lagrangian for the gravitational field beyond the $\Lambda$CDM cosmological model. Ref. [143] obtained cosmological constraints within the Brans–Dicke theory considering how the evolution of the gravitational constant $G$, contained in $\chi$, affects the SNe Ia peak luminosity. The most natural extended framework is the $f(R)$-gravity proposal [162,163,167,381] which contains only an additional scalar degree of freedom. For instance, ref. [323] try to alleviate the $H_0$ tension considering exponential and power-law $f(R)$ models.

The formulation of the $f(R)$ theories in an equivalent scalar-tensor paradigm turns out to be particularly intriguing for our purposes: the function $f(R)$ is restated as a real scalar field $\phi$, which is non-minimally coupled to the metric in the JF. The information about the function $f$ turns into the expression of the scalar field potential $V(\phi)$. The relevance of modified gravity models relies on the possibility that this revised scenario for the gravitational field can account for the physics of the so-called "dark universe" component without the need for a cosmological constant. Indeed, the observed cosmic acceleration in the late universe via the SNe Ia data is a pure dynamical effect, i.e., associated with a modification of the Einsteinian gravity at very large scales (in the order of the present Hubble length).

According to the standard literature on this field (which includes a large number of proposals), three specific $f(R)$ models, i.e., the Hu–Sawicki [382], the Starobinsky [383], and Tsujikawa models [384,385], successfully describe the Dark Energy component (say an effective parameter for the Dark Energy $w = w(z) < -1/3$) and overcome all local constraints. The difference in the form of the Lagrangian densities associated with $f(R)$ models is reflected in the morphology of the potential term governing the dynamics of the scalar field. For instance, the scalar field potential related to the Hu–Sawicki $f(R)$ proposal, with the power index $n = 1$, in the JF is given by

$$V(\phi) = \frac{m^2}{c_2}\left[c_1 + 1 - \phi - 2\sqrt{c_1(1-\phi)}\right], \tag{18}$$

where we have two free parameters $c_1$ and $c_2$, while $m^2 = \chi\rho_{0m}/3$. The scalar-tensor dynamics in the JF for a flat FLRW metric with a matter component is summarized by

$$H^2 = \frac{\chi\rho}{3\phi} - H\frac{\dot{\phi}}{\phi} + \frac{V(\phi)}{6\phi} \tag{19}$$

$$\frac{\ddot{a}}{a} = -\frac{\chi\rho}{3\phi} - \frac{V(\phi)}{6\phi} + \frac{1}{6}\frac{dV}{d\phi} + \frac{\dot{a}\dot{\phi}}{a\phi} \tag{20}$$

$$3\ddot{\phi} - 2V(\phi) + \phi\frac{dV}{d\phi} + 9H\dot{\phi} = \chi\rho, \tag{21}$$

which are the generalized Friedmann equation, the generalized cosmic acceleration equation and the scalar field equation, respectively [167]. We recall that $\phi = \phi(t)$ is a function of the time (or the redshift $z$) only for an isotropic universe. Considering the first term on the right-hand side of Equation (19), it is possible to recognize that $\phi$ mediates the

gravity-matter coupling, and therefore it mimics a space-time varying Einstein constant. Hence, to account for our observed decay of $H_0(z)$, we have to require that the scalar field assumes a specific behavior with the redshift, i.e.,

$$\phi(z) = (1+z)^{2\eta}. \tag{22}$$

Moreover, the remaining terms contained in the gravitational field equations must be negligible. This situation is naturally reached when the potential term is sufficiently slow-varying in a given time interval. We see that the hypothesis of a near-frozen scalar field evolution is a possible assumption, as far as the potential term should provide a dynamical impact, sufficiently close to a cosmological constant term. These simple considerations lead us to claim that this scenario is worth to be investigated for the behavior of $H_0(z)$ here observed.

The specific cosmological models affect the expression of the luminosity distance and this should be the starting point of a careful test of a $f(R)$ theory versus the comprehension of the $H_0$ tension. A new binned analysis of the PS, using the corrected luminosity distance obtained through a reliable $f(R)$, may in principle shed new light on the observed decaying trend of $H_0(z)$, testing also new physics. This analysis is performed in the next Section.

As a preliminary approach, we try to understand which profile we could expect for the scalar field potential, inferred from the behavior of $H_0(z)$. This is quite different from a standard analysis of $f(R)$ models. Generally, a specific $f(R)$ function is defined a priori, and then the dynamical equations are studied to obtain constraints on the free parameters. Here, instead, starting from the observed decreasing trend of $H_0(z)$ and assuming $\phi(z)$ from Equation (22), we wonder what the scalar field potential would be in a scalar-tensor dynamics. Eventually, we should have a scalar field in near-frozen dynamics, i.e., a slow-roll of the scalar field potential, mimicking a cosmological constant term ($\phi \to 1$). To this end, we rewrite the generalized Friedmann Equation (19) and calculate $V(\phi)$:

$$V(\phi) = 6(1-2\eta)\left(\frac{dz}{dt}\right)^2 \phi^{1-1/\eta} - 6m^2 \phi^{3/2\eta}, \tag{23}$$

where we have used the standard definition of redshift and the relation (22) for $\phi(z)$. Moreover, we recall that for a matter component $\rho \sim (1+z)^3$. As a final step, we need to calculate the term $\frac{dz}{dt}$. Starting again from the redshift definition, it is well known that

$$\frac{dz}{dt} = -(1+z)\,H(z). \tag{24}$$

In principle, we would need to compute the Hubble parameter $H(z)$ from the field equations, and then replace $H(z)$ in the term $\frac{dz}{dt}$. However, this procedure is not viable, since we need to fix a well-defined $V(\phi)$ to solve the field equations. Moreover, $H(z)$ appears also in the right-hand-side of Equation (19), because of the non-minimal coupling with the scalar field. Therefore, we can not calculate exactly $\frac{dz}{dt}$ to get $V(\phi)$ in the JF.

Then, to obtain $V(\phi)$ inferred from the trend of $H_0(z)$, we require that the Hubble function provides the same physical mechanism suggested from our binned analysis in Section 4, i.e., simply replacing $H_0$ with $H_0(z)$ given by Equation (9) in the standard Friedmann equation in the $\Lambda$CDM model. With this new definition of $\mathcal{H}_0$, we write the following condition on the Hubble function:

$$H(z) = \frac{\mathcal{H}_0}{(1+z)^\eta}\sqrt{\Omega_{0m}(1+z)^3 + 1 - \Omega_{0m}}. \tag{25}$$

In doing so, using Equations (23)–(25), we determine the form of the scalar field potential

$$\frac{V(\phi)}{m^2} = 6(1-2\eta)\left(\frac{1-\Omega_{0m}}{\Omega_{0m}}\right) - 12\eta\,\phi^{3/2\eta} \tag{26}$$

inferred from the decreasing trend of $H_0(z)$. In other words, the potential Equation (26) might provide an effective Hubble constant that evolves with redshift. In the computation, we have used the expression $\Omega_{0m} = m^2/\mathcal{H}_0^2$.

In Figure 3, we plotted this potential profile, observing that, as expected, a flat region consistently appears, validating our guess on the feasibility of $f(R)$-gravity in the JF to account for the observed behavior of $H_0(z)$. We set $\eta = 0.009$ in Figure 3, according to our binned analysis results for three bins (see Table 1). We stress that the flatness of the potential does not emerge throughout the Pantheon sample redshift range, $0 < z < 2.3$, but it appears only in a narrow region for $0 < z \lesssim z^*$, where $z^* = 0.3$ is the redshift at the Dark Energy and Matter components equivalence of the universe. This form of $V(\phi)$ is reasonable since the Dark Energy contribution, provided by the scalar field in the JF gravity, dominates the matter component only for $0 < z \ll z^*$. It is the weak dependence of $H_0$ on $z$ that ensures the existence of a flat region of the potential, according to the theoretical scenario argued above.

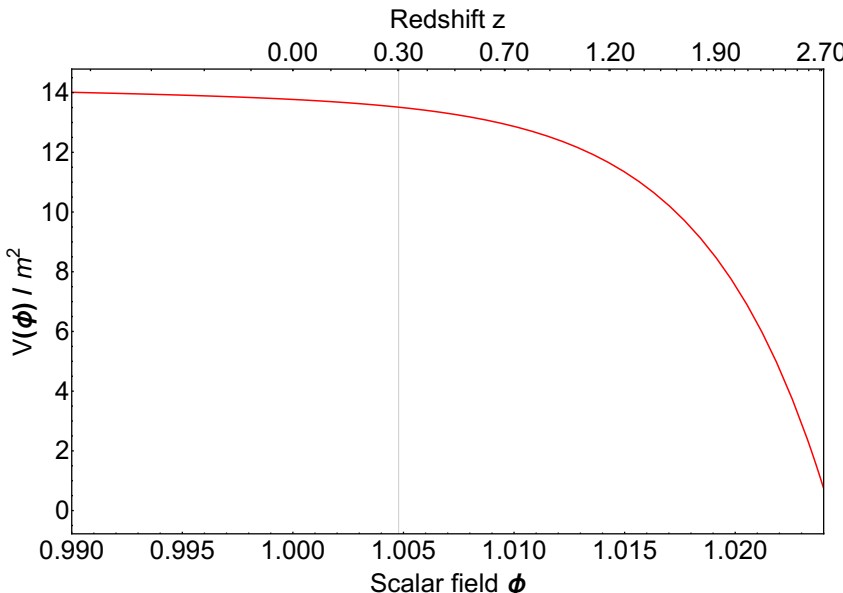

**Figure 3.** Profile of the scalar field potential $V(\phi)$ in the JF equivalent scalar-tensor formalism of the f(R) modified gravity. The form of $V(\phi)$) is inferred from the behavior of $H_0(z)$ (Equation (9)). Note that $V(\phi)/m^2$ is a dimensionless quantity. A flat profile of $V(\phi)$ occurs only at low redshifts, for $0 < z \lesssim 0.3$ or equivalently $\phi \lesssim 1.005$. Note, also, the non-linearity of the scale for the redshift axis on top, considering the relation (22) between $\phi$ and $z$. In this plot, $\eta = 0.009$.

Finally, we can calculate the form of the $f(R)$ function associated with the potential profile. Recalling the following general relations in the JF [167]:

$$R = \frac{dV}{d\phi}, \tag{27}$$

$$f(R) = R\,\phi(R) - V(\phi(R)), \tag{28}$$

we can obtain:

$$f(R) = -6\,m^2 \left[ (1 - 2\eta)\frac{1 - \Omega_{0m}}{\Omega_{0m}} + (3 - 2\eta)\left(-\frac{R}{18\,m^2}\right)^{\frac{3}{3-2\eta}} \right]. \tag{29}$$

Note that the formula above provides a generalization of the Einstein theory of gravity, as it should be in the context of a $f(R)$ model. Indeed, if $\eta = 0$, then $f(R) \equiv R$ reproduces exactly the Einstein–Hilbert Lagrangian density in GR with a cosmological constant $\Lambda$, as soon

as you recognize that $\Lambda = 3m^2(1 - \Omega_{0m}) / \Omega_{0m}$ for a flat geometry, using $m^2 = \mathcal{H}_0^2 \Omega_{0m}$. In particular, expanding the function (29) for $\eta \sim 0$, we can see explicitly the deviation from the Einstein–Hilbert term:

$$f(R) \approx \left( R - 6\, m^2\, \frac{1 - \Omega_{0m}}{\Omega_{0m}} \right) + \frac{2}{3}\eta \left[ R \ln\left( -\frac{R}{m^2} \right) - (1 + \ln 18)\, R + 18m^2\, \frac{1 - \Omega_{0m}}{\Omega_{0m}} \right] + O\left( \eta^2 \right). \tag{30}$$

The first term at the zero-th order in $\eta$ is exactly the Einstein–Hilbert Lagrangian density, while the linear term in $\eta$ provides the correction to GR. Therefore, $\eta$, in addition to being the physical parameter that describes the evolution of $H_0(z)$, also denotes the deviation from GR and the standard cosmological model. It is worthwhile to remark that the expression above may not be the final form of the underlying modified theory of gravity, associated with the global universe dynamics, but only its asymptotic form in the late Universe, i.e., as the scalar of curvature approaches the value corresponding to the cosmological constant in the ΛCDM model. In all these computations we do not consider relativistic or radiation components at very high redshifts, but it may be interesting to test this model with other local probes in the late Universe.

In this discussion, we infer that the dependence of $H_0$ on the redshift points out the necessity of new physics in the description of the universe dynamics and that such a new framework may be identified in the modified gravity, related to metric theories.

## 7. The Binned Analysis with Modified f(R) Gravity

To try to explain the observed trend of $H_0(z)$, we focus on $f(R)$ theories of gravity, and then we perform the same binned analysis, using the correction for the distance luminosity according to the modified gravity. We start from the gravitational field action [167]:

$$S_g = \frac{1}{2\chi} \int d^4x \sqrt{-g} f(R), \tag{31}$$

where $f(R)$, as a function of the Ricci scalar $R$, is an extra degree of freedom compared to General Relativity. We rewrite $f(R) = R + F(R)$ to highlight the deviation from the standard gravity. Varying the total action with respect to the metric, we obtain the flat FLRW metric field equations:

$$H^2(1 + F_R) = \frac{\chi\rho}{3} + \left[ \frac{R\, F_R - F}{6} - F^{RR} H \dot{R} \right], \tag{32}$$

where $F_R \equiv \frac{dF(R)}{dR}$. The Ricci scalar $R$ can be cast in the form

$$R = 12H^2 + 6HH', \tag{33}$$

where the Hubble parameter $H$ is expressed as a function of $\gamma \equiv ln(a)$, and the prime indicates the derivative with respect to $\gamma$.

Now, we introduce two dimensionless variables [382]

$$y_H = \frac{H^2}{m^2} - \frac{1}{a^3}, \qquad\qquad y_R = \frac{R}{m^2} - \frac{3}{a^3}, \tag{34}$$

which denote the deviation of $H^2$ and $R$ with respect to the matter contribution when compared to the ΛCDM model. We rewrite the modified Friedmann Equation (32) and the

Ricci scalar relation (33) in terms of $y_H$ and $y_R$. Then, we have a set of coupled ordinary differential equations:

$$y'_H = \frac{1}{3}y_R - 4y_H \tag{35}$$

$$y'_R = \frac{9}{a^3} - \frac{1}{y_H + a^{-3}}\frac{1}{m^2 F_{RR}}\left[y_H - F_R\left(\frac{1}{6}y_R - y_H - \frac{a^{-3}}{2}\right) + \frac{1}{6}\frac{F}{m^2}\right]. \tag{36}$$

The solution of this coupled first-order differential equations system above can not be obtained analytically, but can be numerically calculated. We need initial conditions such that this scenario mimics the $\Lambda$CDM model in the matter dominated universe at initial redshift $z_i \gg z^*$. Hence, we impose the following conditions for $y_H$ and $y_R$ at the redshift $z_i$:

$$y_H(z_i) = \frac{\Omega_{0\Lambda}}{\Omega_{0m}} \tag{37}$$

$$y_R(z_i) = 12\frac{\Omega_{0\Lambda}}{\Omega_{0m}}. \tag{38}$$

The standard $\Lambda$CDM model is reached for $z = z_i$ or asymptotically, and we consider a flat geometry, such that $\Omega_{0\Lambda} = 1 - \Omega_{0m}$. Finally, the luminosity distance can be written as

$$d_L(z) = \frac{(1+z)}{H_0}\int_0^z \frac{dz'}{\sqrt{\Omega_{0m}\left(y_H(z') + (1+z')^3\right)}}, \tag{39}$$

including the solution $y_H(z)$ from Equation (34) [386].

*Hu–Sawicki Model*

We focus on the Hu–Sawicki model with $n = 1$, considering a late-time gravity modification, described by the following function [382]:

$$f(R) = R + F(R) = R - m^2 \frac{c_1(R/m^2)^n}{c_2(R/m^2)^n + 1}, \tag{40}$$

corresponding to the potential $V(\phi)$ in Equation (18). The parameters $c_1$ and $c_2$ are fixed by the following conditions [382]

$$\frac{c_1}{c_2} \approx 6\frac{\Omega_{0\Lambda}}{\Omega_{0m}} \tag{41}$$

$$F_{R0} \approx -\frac{c_1}{c_2^2}\left(\frac{12}{\Omega_{0m}} - 9\right)^{-2}, \tag{42}$$

where $F_{R0}$ is the value of the field $F_R \equiv dF/dR$ at the present time, and $F(R)$ is the deviation from the Einstein–Hilbert Lagrangian density. Cosmological constraints provide $|F_{R0}| \leq 10^{-7}$ from gravitational lensing and $|F_{R0}| \leq 10^{-3}$ from Solar system [387,388]. We explore several choices of $F_{R0}$.

To simplify the numerical integration of the modified luminosity distance (39), we approximate the numerical solution $y_H$, obtained from the system (35), by a polynomial of order 8. This function is an accurate representation of $y_H$ when we restrict the solution to the range of PS (see Figure 4).

As a consequence, we obtain constraints on $c_1$ and $c_2$, according to Equations (41) and (42). Then, we perform the same binned analysis of Section 4 using the Hu–Sawicki model and the modified luminosity distance (39).

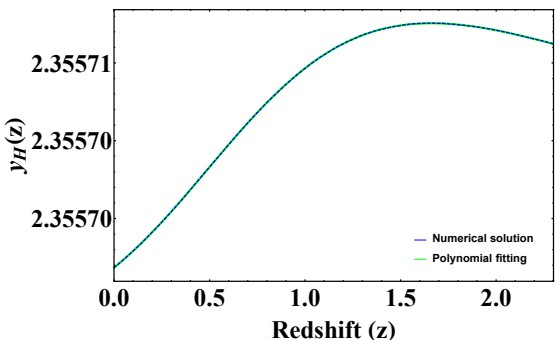

**Figure 4.** The numerical solution for Equation (35) (blue dashed curve) plotted together with its polynomial fitting (green continuous curve) in the case of $F_{R0} = -10^{-7}$. The assumption of a function of redshift in the form of a order-8 polynomial allows an accurate fit for the numerical values. The same fitting procedure has been used in the $F_{R0} = -10^{-4}$ case.

We here run the analysis both for the case of $\Omega_{0m}$ fixed to a fiducial value of 0.298 and for several values of $F_{R0} = -10^{-7}, -10^{-6}, -10^{-5}, -10^{-4}$ (see Table 2 and Figure 5) or we let $\Omega_{0m}$ vary with the two values of $F_{R0} = -10^{-7}, -10^{-4}$ (see Table 3 and Figure 6) for the SNe alone and with SNe +BAOs. Note also that the $\eta$ parameters are all consistent for the several values of $F_{R0}$ in 1 $\sigma$, as you can see in Table 2, for both SNe Ia and SNe Ia + BAOs. Moreover, the values of $\eta$ are consistent in 1 $\sigma$ with the ones obtained from the analysis of the $\Lambda$CDM model (see also Table 1). We consider the cases $F_{R0} = -10^{-7}$, $F_{R0} = -10^{-4}$ and, to study how these results may vary according to the different values of $\Omega_{0m}$ chosen, we tested the model with four values of $\Omega_{0m} = (0.301, 0.303, 0.305)$ taken from the 1 $\sigma$ from a Gaussian distribution centred around the most probable value of 0.298, see [368].

We show in Figure 6 the comparison between the different applications of the Hu–Sawicki model: in the left panels (upper and lower), we consider SNe Ia only, while in the right panels (upper and lower) we combine SNe Ia+BAOs. We here remind that the assumed values for $|F_{R0}|$ of $10^{-4}$ and $10^{-7}$ are well constrained by the $f(T)$ theories [167,382].

**Table 2.** Fitting parameters of $H_0(z)$ for three bins within the Hu–Sawicki model, with SNe only and SNe + BAOs with a fixed value of $\Omega_{0m} = 0.298$ and with several values of $F_{R0}$: $-10^{-4}, -10^{-5}, -10^{-6}, -10^{-7}$. The columns contains: (1) the number of bins; (2) $\mathcal{H}_0$, (3) is $\eta$, according to Equation (9); (4) how many $\sigma$s $\eta$ is compatible with zero (namely, the ratio $\eta/\sigma_\eta$); (5) $F_{R0}$ values; (6) the sample used.

| Hu–Sawicki Model, Results of the Redshift Binned Analysis | | | | | |
|---|---|---|---|---|---|
| **Bins** | $\mathcal{H}_0$ | $\eta$ | $\frac{\eta}{\sigma_\eta}$ | $F_{R0}$ | **Sample** |
| 3 | $70.089 \pm 0.144$ | $0.008 \pm 0.006$ | 1.2 | $-10^{-4}$ | SNe |
| 3 | $70.127 \pm 0.128$ | $0.008 \pm 0.006$ | 1.4 | $-10^{-4}$ | SNe + BAOs |
| 3 | $70.045 \pm 0.052$ | $0.007 \pm 0.002$ | 3.0 | $-10^{-5}$ | SNe |
| 3 | $70.062 \pm 0.132$ | $0.007 \pm 0.005$ | 1.3 | $-10^{-5}$ | SNe + BAOs |
| 3 | $70.125 \pm 0.046$ | $0.010 \pm 0.002$ | 5.4 | $-10^{-6}$ | SNe |
| 3 | $70.115 \pm 0.153$ | $0.008 \pm 0.007$ | 12.1 | $-10^{-6}$ | SNe + BAOs |
| 3 | $70.118 \pm 0.131$ | $0.011 \pm 0.006$ | 1.9 | $-10^{-7}$ | SNe |
| 3 | $70.053 \pm 0.150$ | $0.007 \pm 0.007$ | 1.1 | $-10^{-7}$ | SNe + BAOs |

Thus, the existence of this trend is, once again, confirmed, and it remains unexplained also in the modified gravity scenario. Indeed, a suitable modified gravity model which would be able to predict the observed trend of $H_0$, would allow observing a flat profile of $H_0(z)$ after a binned analysis. Further analysis must be carried out with other Dark Energy

models or other modified gravity theories to investigate this issue in the future, for instance focusing on the proposed model in Section 6.2.2.

**Table 3.** Fitting parameters of $H_0(z)$ for three bins within the Hu–Sawicki model, with SNe and SNe + BAOs by fixing several values of $\Omega_M = 0.298, 0.303, 0.301, 0.305$ and values of $F_{R0} = -10^{-4}$ and $F_{R0} = -10^{-7}$. The columns are as follows: (1) the $\Omega_{0m}$ value; (2) $\mathcal{H}_0$, (3) $\eta$, according to Equation (9); (4) how many $\sigma$s the evolutionary parameter $\eta$ is compatible with zero (namely, $\eta/\sigma_\eta$); (5) $F_R0$; (6) the sample used.

| | | Hu–Sawicki Model, Results of the 3 Bins Analysis | | | |
|---|---|---|---|---|---|
| $\Omega_{0m}$ | $\mathcal{H}_0$ | $\eta$ | $\frac{\eta}{\sigma_\eta}$ | $F_{R0}$ | Sample |
| 0.298 | $70.140 \pm 0.045$ | $0.011 \pm 0.002$ | 5.1 | $-10^{-7}$ | SNe |
| 0.298 | $70.050 \pm 0.126$ | $0.007 \pm 0.006$ | 1.2 | $-10^{-7}$ | SNe + BAOs |
| 0.303 | $70.088 \pm 0.075$ | $0.012 \pm 0.004$ | 3.0 | $-10^{-7}$ | SNe |
| 0.303 | $70.004 \pm 0.139$ | $0.009 \pm 0.007$ | 1.3 | $-10^{-7}$ | SNe + BAOs |
| 0.301 | $70.054 \pm 0.056$ | $0.009 \pm 0.003$ | 3.0 | $-10^{-7}$ | SNe |
| 0.301 | $70.072 \pm 0.170$ | $0.010 \pm 0.008$ | 1.2 | $-10^{-7}$ | SNe + BAOs |
| 0.305 | $70.048 \pm 0.034$ | $0.012 \pm 0.002$ | 6.0 | $-10^{-7}$ | SNe |
| 0.305 | $70.004 \pm 0.140$ | $0.010 \pm 0.007$ | 1.4 | $-10^{-7}$ | SNe + BAOs |
| 0.298 | $70.135 \pm 0.080$ | $0.009 \pm 0.004$ | 2.2 | $-10^{-4}$ | SNe |
| 0.298 | $70.087 \pm 0.155$ | $0.009 \pm 0.007$ | 1.2 | $-10^{-4}$ | SNe + BAOs |
| 0.303 | $70.096 \pm 0.146$ | $0.012 \pm 0.007$ | 1.7 | $-10^{-4}$ | SNe |
| 0.303 | $70.044 \pm 0.129$ | $0.009 \pm 0.006$ | 1.5 | $-10^{-4}$ | SNe + BAOs |
| 0.301 | $70.111 \pm 0.158$ | $0.012 \pm 0.008$ | 1.5 | $-10^{-4}$ | SNe |
| 0.301 | $70.038 \pm 0.170$ | $0.009 \pm 0.008$ | 1.1 | $-10^{-4}$ | SNe + BAOs |
| 0.305 | $70.074 \pm 0.026$ | $0.016 \pm 0.001$ | 16.0 | $-10^{-4}$ | SNe |
| 0.305 | $70.028 \pm 0.090$ | $0.011 \pm 0.004$ | 2.4 | $-10^{-4}$ | SNe + BAOs |

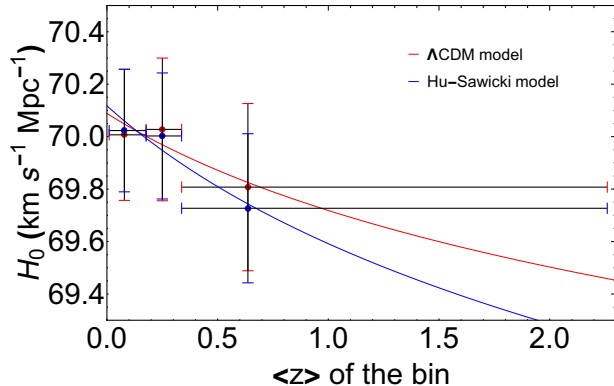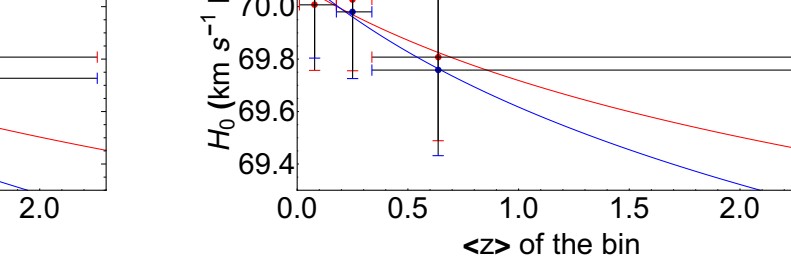

**Figure 5.** *Cont.*

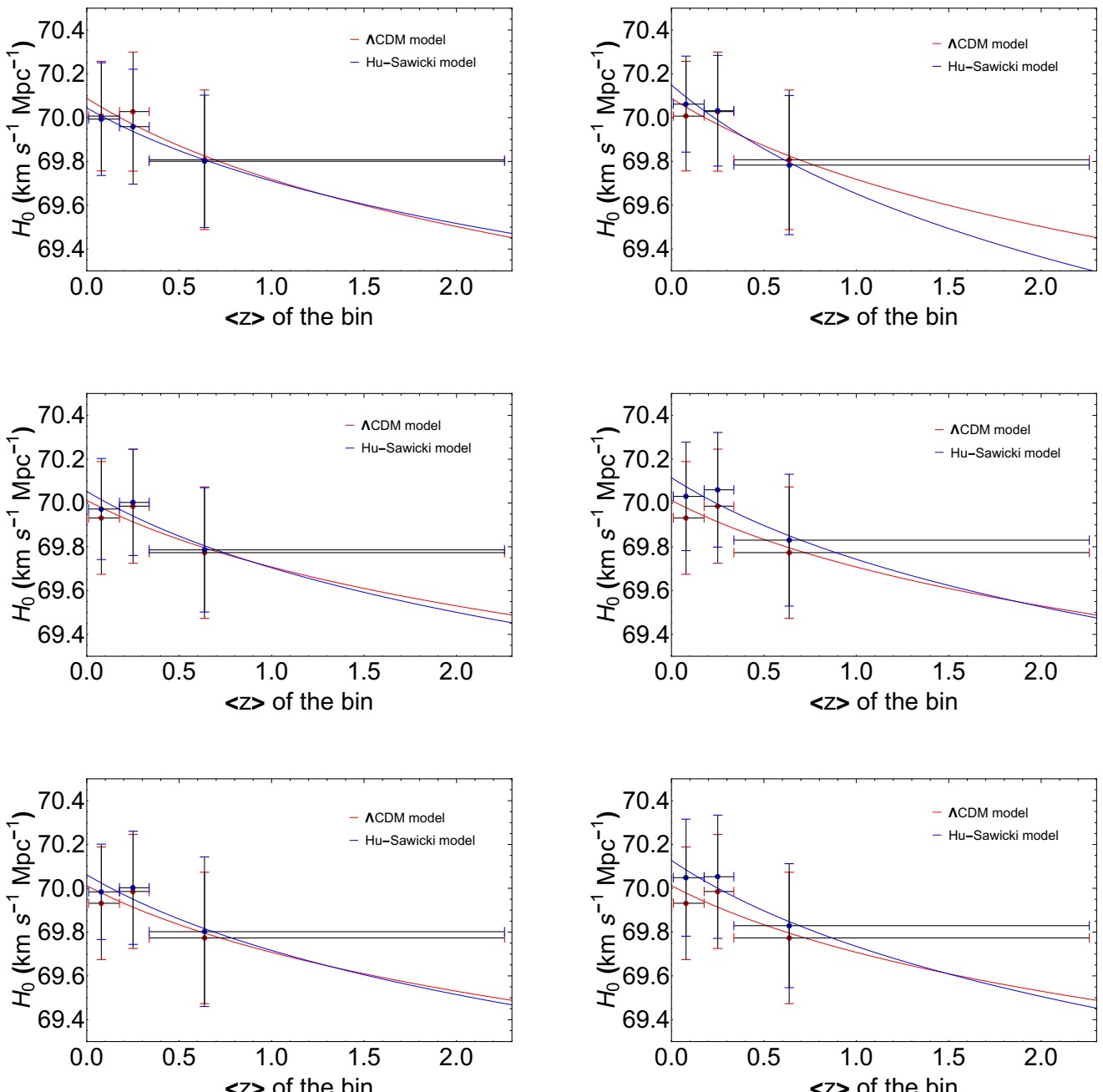

**Figure 5.** The first four panels deal with $H_0$ vs. $z$ for SNe, the four bottom panels include BAO measurements for the H-S model. The upper 4 panels show from the left to the right $F_{R0} = -10^{-7}, -10^{-6}, -10^{-5}, -10^{-4}$, respectively. The standard $\Lambda$CDM cosmology is shown in red and the Hu–Sawicki model in blue. Analogously, the bottom panels have the same notation about the values of $F_{R0}$.

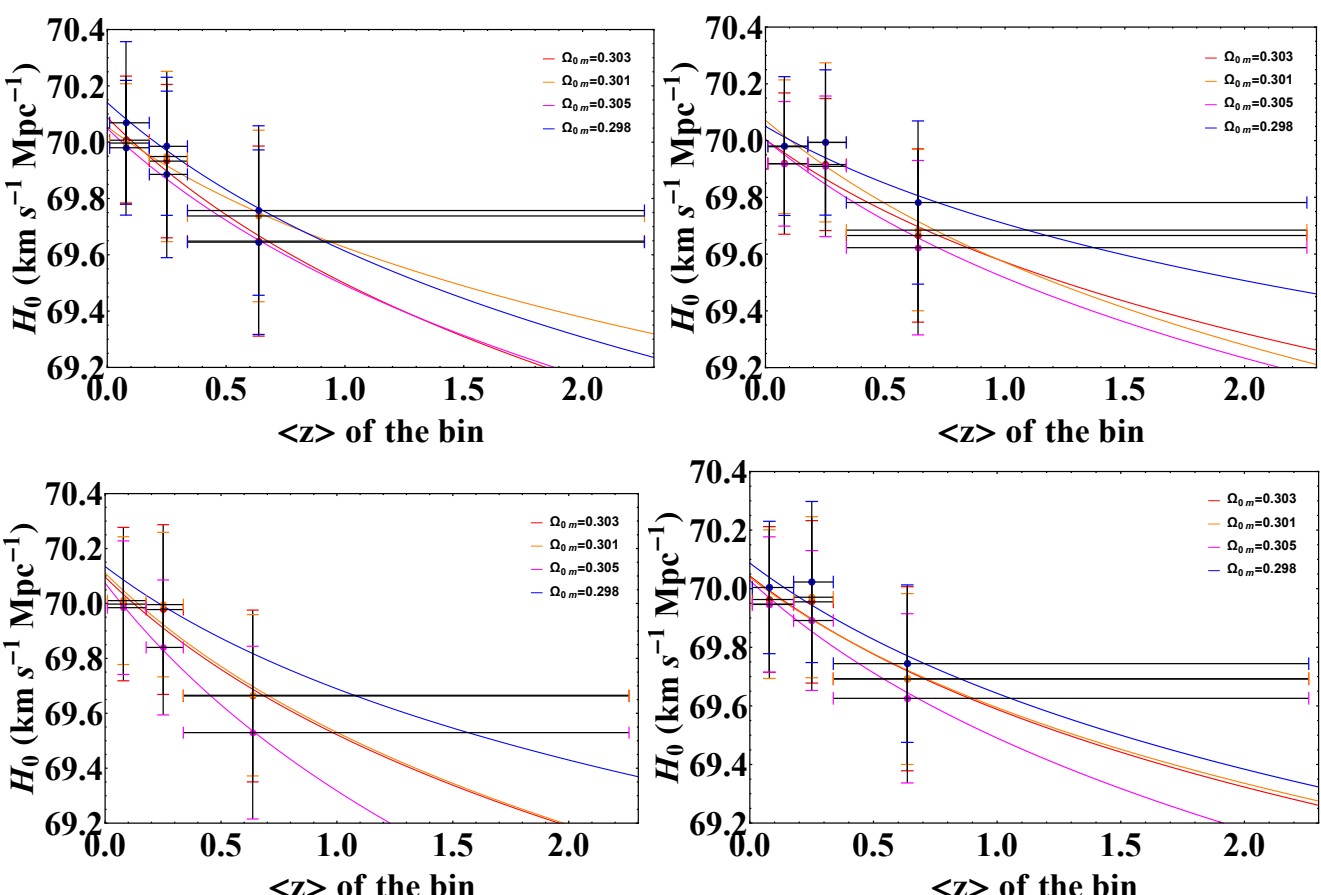

**Figure 6.** The Hubble constant versus redshift plots for the three bins of SNe Ia only, considering the Hu–Sawicki model. **Upper left panel.** The condition of $F_{R0} = -10^{-7}$ is applied to the case of SNe only, with the different values of $\Omega_{0m} = 0.301, 0.303, 0.305$. **Upper right panel.** The same of the upper left, but with the contribution of BAOs. **Lower left panel.** The SNe only case with the $F_{R0} = -10^{-4}$ condition, considering the different values of $\Omega_{0m} = 0.301, 0.303, 0.305$. **Lower right panel.** The same as the lower left, but with the contribution of BAOs. The orange color refers to $\Omega_{0m} = 0.301$, the red to $\Omega_{0m} = 0.303$, the magenta to $\Omega_{0m} = 0.305$, and the blue to $\Omega_{0m} = 0.298$.

## 8. Requirements for a Suitable f(R) Model

Since the Hu–Sawicki model seems to be inadequate to account for the observed phenomenon of the decaying $H_0(z)$, in what follows, we provide some general properties that an $f(R)$ model in the JF must possess to induce the necessary scenario of a slowly varying Einstein constant. Now, we consider again the dynamical impact of the scalar field $\phi$, related to the $f(R)$ function. Let us observe that the following relation holds in the following way:

$$\frac{d\phi}{dz} = -\frac{1}{1+z}\frac{\dot{\phi}}{H}. \tag{43}$$

In order to get the desired behavior $\phi \simeq (1+z)^{2\eta}$, we must deal with a dynamical regime where the following request is satisfied:

$$\frac{\dot{\phi}}{H} = -2\eta\phi. \tag{44}$$

We consider a slow-rolling evolution of the scalar field $\phi$ in the late universe, near enough to $\phi \simeq 1$. Then, we consider in Equation (19) $\rho \sim 0$, because we are in the Dark Energy

dominated phase, and we consider $H_0 \dot{\phi}$ small with respect to the potential term $V(\phi \simeq 1)$. We neglect, also, the term $\ddot{\phi}$. Under these conditions, Equations (19) and (21) become

$$H^2 = \frac{V}{6\phi} \tag{45}$$

and

$$\frac{\dot{\phi}}{H} = \frac{1}{9H^2}\left(2V - \phi\frac{dV}{d\phi}\right), \tag{46}$$

respectively.

Referring to Equation (45) at $z \sim 0$, we make the identification $H_0^2 \equiv V(\phi \simeq 1)/6$. Hence, in order to reproduce Equation (44), we must require that for $\phi \to 1$, the following relation holds:

$$\eta = \frac{1}{3V}\left(\phi\frac{dV}{d\phi} - 2V\right). \tag{47}$$

The analysis above states the general features that a $f(R)$ model in the JF has to exhibit to provide a viable candidate to reproduce the observed decay behavior of $H_0(z)$ (Equation (36)). We conclude by observing that the picture depicted above relies on the concept of a slow-rolling phase of the scalar field, when it approaches the value $\phi \simeq 1$ and, in this respect, the potential term should have for such value a limiting dynamics, which remains there confined for a sufficiently long phase. It is just in such a limit that we are reproducing a $\Lambda$CDM model, but with the additional feature of a slowly varying Einstein constant. As far as the value of $z$ increases, the deviation of the considered model from General Relativity becomes more important, but this effect is observed mainly in the gravity-matter coupling. In other words, the motion of the photon, as observed in the gravitational lensing, is not directly affected by the considered deviation, since the geodesic trajectories in the space-time do not directly feel the Einstein constant value. This consideration could allow for a large deviation of $\phi$ from the unity that is expected in studies of the photons' propagation.

### 8.1. An Example for Low Redshifts

As a viable example for the Dark Energy dominated Universe (slightly different from the traced above), we consider a potential term (and the associated slow-rolling phase) similar to the one adopted in the so-called chaotic inflation [389,390], i.e.,:

$$V(\phi) = \delta + 6H_0^2\phi^2, \tag{48}$$

where $\delta$ is a positive constant, such that $\delta \ll 6H_0^2$. From Equation (45), we immediately get

$$H^2 \simeq H_0^2\phi \sim H_0^2, \tag{49}$$

where, we recall that we are considering the slow-rolling phase near $\phi \to 1$. Analogously, from Equation (47), we immediately get:

$$\eta \sim -\frac{\delta}{9H_0^2}. \tag{50}$$

The negative value of $\eta$ is coherent with the behavior $H^2 \propto \phi$. Hence, we can reproduce the requested behavior of $\phi(z)$ by properly fixing the value of $\delta$ to get $\eta$ as it comes out from the data analysis of Section 4. Specifically, we get $\delta \sim 10^{-3}H_0^2$ to have $\eta \sim 10^{-2}$. Furthermore, it is easy to check that, for $\phi \to 1$, Equations (44) and (49), we find the relation

$$\ddot{\phi} \sim \mid \eta \mid H_0\dot{\phi} \ll 3H_0\dot{\phi}, \tag{51}$$

which ensures that we are dealing with a real slow-rolling phase.

Finally, we compute the $f(R)$ function corresponding to the potential in Equation (48), recalling the relation (28):

$$f(R) = \frac{R^2}{24H_0^2} - \delta \tag{52}$$

We conclude by observing that this specific model is reliable only as far the universe matter component is negligible, $z < 0.3$. The Einstein constant in front of the matter-energy density $\rho$ would run as $(1+z)^{2\eta}$. The example above confirms that the $f(R)$ gravity in the JF is a possible candidate to account for the observed effect of $H_0(z)$, but the accomplishment of a satisfactory model for the whole $\Lambda$CDM phase requires a significant effort in further investigation, especially accounting for the constraints that observations in the local universe provided for modified gravity.

*8.2. Discussion*

Let us now try to summarize the physical insight that we can get from the analysis above, about the possible theoretical nature of the observed $H_0(z)$ behavior. We can keep as a reliably good starting point the idea that the origin of a modified scaling of the function $H(z)$ with respect to the standard $\Lambda$CDM model can be identified in a slowly varying Einstein constant with the redshift. Furthermore, it is a comparably good assumption to search, in the framework of a scalar-tensor formulation of gravity, the natural explanation for such a varying Einstein constant. As shown in Section 6.2, a scalar-tensor formulation can reproduce the required scaling of the function $H(z)$, which we observe as an $H_0(z)$ behavior in the standard $\Lambda$CDM model. Hence, we naturally explored one of the most interesting and well-motivated formulations of a scalar-tensor theory, namely the $f(R)$ gravity in the JF. In this respect, in Section 6.2.2, we first evaluated the form of the scalar field potential inferred from the observed decreasing trend of $H_0(z)$, and our data analysis suggested a model described in Equations (26) and (29). Then, we investigated if, one of the most reliable models for reproducing the Dark Energy effect with modified gravity, i.e., the Hu–Sawicki proposal, was able to induce the requested luminosity distance to somehow remove the observed effect, thus accounting for its physical nature. The non-positive result of this investigation leads us to explore theoretically the question of reproducing simultaneously the Dark Energy contribution and the observed $H_0(z)$ effect, by a single $f(R)$ model of gravity in the JF. In Section 8, it has been addressed this theoretical question, by establishing the conditions that a modified gravity model has to satisfy to reach the simultaneous aims mentioned above. Finally, we considered a specific model for the late universe, based on a slow-rolling picture for the scalar field near its today value $\phi \simeq 1$. This model was successful in explaining the Dark Energy contribution and the necessary variation of the Einstein constant, but it seems hard to be reconciled with the earlier Universe behavior, when the role of the matter contribution becomes relevant. Thus, based on this systematic analysis, we can conclude that the explanation for $H_0(z)$ is probably to be attributed to modified gravity dynamics, but it appears more natural to separate its effect from the existence of a Dark Energy contribution. In other words, we are led to believe that what we discovered about the SNe Ia+BAOs binned analysis must be regarded as a modified gravity physics of the scalar-tensor type, but leaving on a standard Universe, well represented by a $\Lambda$CDM model a priori.

## 9. Conclusions

We analyzed the PS together with the BAOs in three bins in both the $\Lambda$CDM and $w_0 w_a$CDM models to investigate if an evolutionary trend of $H_0$ persists also with the contribution of BAOs and by varying two parameters contemporaneously with $H_0$ ($\Omega_{0m}$ and $w_a$ for the $\Lambda$CDM and $w_0 w_a$CDM, respectively). The persistence of the trend of $H_0$ as a function of redshift is also shown in the case of the Hu–Sawicki model. We here stress that the main goal of the current analysis is to highlight the reliability of the trend of $H_0(z)$ and not to further constrain $\Omega_{0m}$ or any other cosmological parameters. With the subsequent fitting of $H_0$ values through the model $g(z) = \mathcal{H}_0/(1+z)^\eta$, we obtain

$\eta \sim 10^{-2}$, as in the previous work [51]: those are compatible with zero from 1.2 to 5.8 $\sigma$ (see Table 1). The multidimensional results could reveal a dependence on the redshift of $H_0$, assuming that it is observable at any redshift scale. If this evolution is not caused by statistical effects and other selection biases or hidden evolution of SNe Ia parameters [364], we show how $H_0(z)$ could modify the luminosity distance definition within the modified theory of gravity. If we consider a theoretical interpretation for the observed trend, new cosmological scenarios may explain an evolving Hubble constant with the redshift. For instance, we test in Sections 6.2 and 7 a simple class of modified gravity theories given by the $f(R)$ models in the equivalent scalar-tensor formalism. In principle, this could be due to an effective varying Einstein constant governed by a slow evolution of a scalar field which mediates the gravity-matter interaction. However, the slow decreasing trend of $H_0$ has proven to be independent of the Hu–Sawicki model application. Indeed, if this theory had worked we would have observed the trend of the $\eta$ parameter to be flattened out and be compatible with 0 in 1 $\sigma$ at any redshift bin. This is not the case, thus new scenarios must be explored within the modified theories of gravity or slightly alternative approaches (see Section 8.2). We can state that this evolving trend of $H_0$ is independent of the starting values of the fitting for $H_0$ (we here have considered $H_0 = 70$) and, thus, on the fiducial $M$ and on the redshift bins and even when we consider two cosmological parameters changing contemporaneously ($\Omega_{0m}$ and $w_a$ in $\Lambda$CDM and $w_0 w_a$CDM models, respectively). Thus, we need to further investigate the nature of this trend. In addition, the implementation of GRBs as cosmological probes together with SNe Ia and BAOs has proven to be not only possible in a near future but also necessary since the redshift range that GRBs cover is much larger than the one typical of SNe Ia. This last characteristic will surely allow GRBs to give further information on the nature of the early universe and pose new constraints in the future measurements of $H_0$.

**Author Contributions:** M.G.D. performed the conceptualization of all project, data curation, formal analysis, methodology, writing original draft, validation, supervision, software. B.D.S. performed data curation, visualization, formal analysis, methodology, writing original draft, software. T.S. performed formal analysis, visualization, methodology, writing original draft on f(R) and revised it, G.M. performed a partial conceptualization limited to the theoretical part of the f(R) gravity theory; E.R. edited and review the analysis on f(R) and participated in the general discussion and conceptualization of the paper. G.L. wrote Section 6.2.1 and gave suggestions on the cosmological constraints on $w$. M.B. performed the conceptualization on the priors to answer the referee report. S.U. performed a formal analysis on changing the parameters together with $H_0$. All authors have read and agreed to the published version of the manuscript.

**Funding:** This research received no external funding.

**Institutional Review Board Statement:** Not applicable.

**Informed Consent Statement:** Not applicable.

**Data Availability Statement:** Not applicable.

**Acknowledgments:** This work made use of Pantheon sample data [284], which can be found in the GitHub repository: https://github.com/dscolnic/Pantheon (accessed on 21 December 2020). This work made use of data supplied by the UK Swift Science Data Centre at the University of Leicester. We are thankful to V. Nielson, A. Lenart, G. Sarracino, and D. Jyoti for their support on cosmological computations. T. Schiavone is supported in part by INFN under the program TAsP (Theoretical Astroparticle Physics). G.L. and B.D.S. acknowledge the support of INFN. T.S. acknowledges the support of the Department of Physics of the University of PISA. M.G.D. acknowledges the support from NAOJ and NAOJ—Division of Science.

**Conflicts of Interest:** The authors declare no conflict of interest.

## Notes

1.    https://github.com/dscolnic/Pantheon (accessed on 21 December 2020).
2.    The code is available upon request.

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
