# Peer review of "On the Evolution of the Hubble Constant with the SNe Ia Pantheon Sample and Baryon Acoustic Oscillations: A Feasibility Study for GRB-Cosmology in 2030"

_galaxies, doi:10.3390/galaxies10010024_

Round 1

Reviewer 1 Report

This paper is meant to be a follow up of a previous work published by the authors themselves, as correctly reported in reference [1]. The novel content of this manuscript consists of two aspects: i) the improvement of the analysis done in [1] by considering also other cosmological data such as BAO and ii) a theoretical discussion about the possible explanation for their results.

In particular, the authors claim that this new analysis seems to confirm the decreasing trend of Hubble constant throughout the investigated samples n redshift and some good candidates for explaining this trend are discussed both from the astrophysical and cosmological point of view. As an overall judgement, I think that the paper is well written and the results are presented clearly and explained in details. However, few concerns of mine must be addressed before I can recommend to paper for publication in Galaxies. In particular:

1) In regard of the analysis within the Cosmic Concordance Model, the authors allow only H0 to vary as a free parameters and the assume that Omega matter is fixed to some fiducial value. I wonder what happens if the value of Omega matter is allowed to vary. Indeed, a discussion about what happens when a marginalization over Omega matter is needed, since their deviation from LambdaCDM model is significant only within 2 sigmas.

2) The same comment as before should be done also the subsequent analysis regarding other cosmological models. In general, I strongly recommend the authors discuss model by model what happens when all the parameters, such as w0, wa, etc..., are allowed to vary along the fit over the sample. This could shade some light about the robustness of their results.

3) For what concerns the model presented in Sect. 7, I would recommend the authors to clarify the way Omega Lambda is treated: is there any prior linking it to Omega matter or is it just fixed to a fiducial value? I would expect, indeed, that a relation between Omega Lambda and Omega matter is still present, even though it might be not so simple as the Lambda CMD one. Again, also in this case, a discussion about what happens when all the parameters are allowed to vary should be in order.

4) Even though the reference list looks exhaustive, I think that a mention to the theoretical works about the study of the Hubble diagram within either perturbation theory or numerical studies can be improved. In this regards, I would suggest the authors to have a look at these papers:

  • JCAP 06 (2013) 002 arXiv:1302.0740 [astro-ph.CO]
  • JCAP 03 (2017) 062 arXiv:1612.03726 [astro-ph.CO]
  • Phys.Rev.D 100 (2019) 2, 021301 arXiv:1812.04336 [astro-ph.CO]
  • JCAP 02 (2020) 017 arXiv:1911.09469 [gr-qc]

Author Response

Please look at the pdf below

Reviewer 2 Report

Overall manuscript is good, but conclusion should be improved.

Author Response

Below it is the new version of the manuscript

Reviewer 3 Report

In this article the authors have studied "On the evolution of the Hubble constant with the SNe Ia Pantheon Sample and Baryon Acoustic Oscillations". My comments are appended below:

  1. The overall idea of the paper is excellent and the authors have presented in a very well fashioned way.
  2. The numerical outcomes, plots and the analysis part of the paper are very nice and discussed very elaborately in the paper.
  3. Reference list is complete.

Author Response

Thanks for review.

Reviewer 4 Report

The paper may be published as it is, though the introduction is a bit disproportionate with respect to the paper. A vast portion of the long list of references is not directly related to the work exposed in the paper.

Author Response

Thanks for review.

Round 2

Reviewer 1 Report

This version of the manuscript have addressed all my previous concerns. The authors have improved the analysis following my comments and properly explained the strategy adopted for such a purpose and its limits. I can then recommend the paper in its present form for publication in Galaxies. Congratulations to the authors.